



Geoscientific
Model Development

# A model for marine sedimentary carbonate diagenesis and paleoclimate proxy signal tracking: IMP v1.0

**Yoshiki Kanzaki**[1,a]**, Dominik Hülse**[1]**, Sandra Kirtland Turner**[1]**, and Andy Ridgwell**[1]

[1]Department of Earth and Planetary Sciences, University of California – Riverside, Riverside, CA 92521, USA
[a]currently at: School of Earth and Atmospheric Sciences, Georgia Institute of Technology, Atlanta, GA 30332, USA

**Correspondence:** Yoshiki Kanzaki (ykanzaki3@gatech.edu)

**Abstract.** The preservation of calcium carbonate in marine sediments is central to controlling the alkalinity balance of the ocean and, hence, the ocean–atmosphere partitioning of $CO_2$. To successfully address carbon cycle–climate dynamics on geologic ($\gg 1$ kyr) timescales, Earth system models then require an appropriate representation of the primary controls on $CaCO_3$ preservation. At the same time, marine sedimentary carbonates represent a major archive of Earth history, as they have the potential to preserve how seawater chemistry, isotopic composition, and even properties of planktic and benthic ecosystems, change with time. However, changes in preservation and even chemical erosion of previously deposited $CaCO_3$, along with the biogenic reworking of upper portions of sediments, whereby sediment particles are translocated both locally and nonlocally between different depths in the sediments, all act to distort the recorded signal. Numerical models can aid in recovering what the "true" environmental changes might have been, but only if they appropriately account for these processes.

Building on a classical 1-D reaction-transport framework, we present a new diagenetic model – IMP (Implicit model of Multiple Particles (and diagenesis)) – that simulates biogeochemical transformations in carbonate-hosted proxy signals by allowing for populations of solid carbonate particles to possess different physicochemical characteristics such as isotopic value, solubility and particle size. The model also utilizes a variable transition matrix to implement different styles of bioturbation. We illustrate the utility of the model for deciphering past environmental changes using several hypothesized transitions of seawater proxies obscured by sediment mixing and chemical erosion. To facilitate the use of IMP, we provide the model in Fortran, MATLAB and Python

versions. We described IMP with integration into Earth system models in mind, and we present the description of this coupling of IMP with the "cGENIE.muffin" model in a subsequent paper.

## 1 Introduction

The removal of carbon and alkalinity through the preservation and burial of carbonate minerals in accumulating marine sediments plays a central role in the global carbon cycle and, hence, the regulation of climate over geologic timescales (e.g., Ridgwell and Zeebe, 2005; Kump et al., 2009). Specifically, burial of $CaCO_3$ is the major long-term sink for atmospheric $CO_2$ ($> \sim 10^4$ years), while chemical erosion of $CaCO_3$ works as a buffer against short-term ($\sim 10^2$ to $10^4$ years) ocean acidification that accompanies $CO_2$ emissions (e.g., Broecker and Takahashi, 1977; Berner et al., 1983; Archer et al., 1998; Ridgwell and Zeebe, 2005). As such, the dynamics of the calcium carbonate cycle are also important to the stability of the marine environment inhabited by calcifying (and carbonate chemistry sensitive) organisms such as corals (Hönisch et al., 2012) and takes on particular importance in the context of the release of $CO_2$ to the ocean–atmosphere system, both past and present or future (e.g., Archer et al., 1997, 1998; Zeebe and Zachos, 2007; Boudreau et al., 2010; Lord et al., 2016; Penman et al., 2016).

Although calcium carbonate can be produced diagenetically within the sediments (which we do not address in this initial version of the model and will not discuss in any detail in this paper), $CaCO_3$ is predominantly delivered to ocean sediments from calcifying organisms (principally plankton)

living in the overlying ocean surface, with a minor contribution from organisms living at or close to the sediment surface itself. Two polymorphs exist – calcite (trigonal), which is precipitated by foraminifera and coccolithophores, and aragonite (orthorhombic), which is precipitated by organisms such as modern corals and pteropods. Deep-sea sediments and, hence, marine archives are generally dominated by the calcitic form (although our model is designed to be sufficiently flexible to consider a mix of polymorphs). The crystal structure of $CaCO_3$ allows for the substitution of a variety of trace elements, which along with measurable isotopic properties of most of these elements, serves as an important archive of paleoceanographic proxies. For example, the $\delta^{13}C$ record of $CaCO_3$ has been widely used to constrain C transfer between reservoirs (e.g., Kump and Arthur, 1999), the $\delta^{18}O$ record to reconstruct past water temperature and/or global ice volume (e.g., Zachos et al., 2001; Dunkley Jones et al., 2013), the $\delta^{11}B$ record for paleo-ocean pH reconstruction (e.g., Gutjahr et al., 2017), and I/Ca ratios to estimate the ocean redox state in the past (e.g., Lu et al., 2018). However, reconstruction of paleoenvironments using $CaCO_3$-based proxies is complicated by $CaCO_3$ loss via dissolution (chemical erosion) and mixing of $CaCO_3$ particles within sediments by benthic organisms (bioturbation). Both phenomena are ubiquitous and need to be accounted for when one reads proxies in sedimentary carbonates, particularly for events that occur rapidly relative to the sediment accumulation timescale (e.g., Bard et al., 1987; Ridgwell, 2007b; Trauth, 2013).

The effect of bio-mixing on the preservation of proxy signals has been examined analytically and numerically depending on the complexity with which sediment bioturbation is represented (e.g., Berger et al., 1977; Bard et al., 1987; Trauth, 1998, 2013; Hull et al., 2011; Steiner et al., 2016; Kirtland Turner et al., 2017). Most of these studies assume either random mixing or diffusion that follows Fick's law (biodiffusion) for bioturbation. Particle mixing by benthos, however, can be more complex than can be captured by biodiffusion or random mixing, as it depends on animal-specific properties such as burrow geometry and feeding rates and styles (e.g., Meysman et al., 2006; Kristensen et al., 2012). For example, Boudreau and Imboden (1987) suggested, based on their analytical examination of the effect of nonlocal mixing on distributions of radiotracers, that animal-specific mixing can result in different sediment particle distributions over time than simple biodiffusion. Therefore, specific, more complex animal behaviors and the resulting bio-mixing need to be simulated with a transition matrix method (e.g., Shull, 2001) or a process-based particle-tracking model such as a lattice–automaton bioturbation simulator (LABS; Boudreau et al., 2001; Choi et al., 2002; Kanzaki et al., 2019). Specific animal behaviors can be reflected by probabilities in the transition matrix or as automaton rules in LABS. Other (more common) models simply employ a biodiffusion coefficient and consider only bulk properties (e.g.,

Ridgwell, 2007a, b), further simplifying how proxy signals are recorded.

Chemical erosion is also known to distort proxy signals (e.g., Keir, 1984; Keir and Michel, 1993; Broecker et al., 1991; Oxburgh, 1998; Barker et al., 2007; Ridgwell, 2007b; Jennions et al., 2015). Moreover, it has been shown that the extent of signal distortion by chemical erosion is related to the strength of biodiffusion (e.g., Keir, 1984). Generally, however, examination of the effect of chemical erosion on proxy signals has been relatively limited compared with that of bioturbation. Most previous studies have focused on explaining older $^{14}C$ ages in sedimentary $CaCO_3$ that suffers more significant dissolution (Keir, 1984; Keir and Michel, 1993; Broecker et al., 1991; Oxburgh and Broecker, 1993; Oxburgh, 1998; Barker et al., 2007), and the models used therein cannot be directly applied to other proxies. Only a limited number of studies have quantitatively discussed the effect of dissolution on other proxy signals (e.g., $\delta^{13}C$ by Jennions et al., 2015). The reason for this is that published sediment mixing models do not generally account for diagenetic reactions (e.g., Trauth, 2013), and even those that enable $CaCO_3$ dissolution are too specific regarding the tracked proxy and style of bioturbation and are, thus, inapplicable to a variety of proxies or to different styles of bioturbation (e.g., Keir, 1984).

Caution is particularly warranted in the interpretation of $CaCO_3$-hosted proxy records during episodes of ocean acidification when both chemical erosion (e.g., Zachos et al., 2005) and changes in benthic ecology and, hence, bioturbation (e.g., Jennions et al., 2015) are expected, such as during hyperthermal events in the early Cenozoic (e.g., Ridgwell, 2007b; Sluijs et al., 2007; McInerney and Wing, 2011). Currently, no model exists that is specifically designed to simulate $CaCO_3$ diagenesis along with different styles of bioturbation, while simultaneously tracking a variety of proxy signals, and hence explicitly tackle complex past geochemical–biological sediment proxy questions.

Here, we present the "Implicit model of Multiple Particles (and diagenesis)" – IMP – that can be used to explore the consequences of chemical erosion and bioturbation on proxy records. IMP is a reactive-transport model of diagenesis for carbonates, organic matter and refractory detrital materials in marine sediments, along with dissolved oxygen and aqueous $CO_2$ species in the porewater. Overlaying this is the ability to track proxy signals in carbonates by representing multiple "classes" of carbonates particles with different proxy values (for more details, see Sect. 2.1). IMP also has the flexibility of representing various styles of solid-phase mixing through the use of different transition matrices. Thus, the model can be used to simulate a wide variety of scenarios of environmental change. Following the presentation of the model framework, we illustrate how the model can be utilized to discern signal distortion caused by chemical erosion and different kinds of bioturbation and to better interpret proxy signals for paleoenvironments.

## 2 Model description

### 2.1 Model overview

IMP builds on the reactive-transport framework of Archer (1991) and, as such, is based on the principals of conservation of carbonate alkalinity and total $CO_2$ in sediment porewater. However, IMP extends the Archer (1991) model to (i) be explicit about depth-dependent and temporal changes in all considered species, (ii) allow more than one "class" of $CaCO_3$ particles (see below for the definition of class) and (iii) simulate a variety of mixing styles caused by bioturbation using transition matrices.

Here, the term $CaCO_3$ class refers to any ensemble of solid $CaCO_3$ particles that (a) record the same proxy value or (b) possess common, distinct biological and physicochemical characteristics. As an example of case (a) above, if two ensembles of $CaCO_3$ particles have distinctive proxy signals (e.g., different $\delta^{13}C$ and/or $\delta^{18}O$ values), we refer to these two ensembles as two distinctive $CaCO_3$ classes, even if they belong to the same model species and have exactly the same geochemical properties (i.e., in a "traditional" reactive-transport framework such as of Archer, 1991, this would all just be "$CaCO_3$"). Similarly (case (b) above), if two ensembles of $CaCO_3$ particles belong to different model species (e.g., have distinct sizes and associated dissolution and bio-mixing properties; Keir, 1980; Walter and Morse, 1984, 1985; Bard, 2001; Schmidt et al., 2004), they are referred to as two distinctive $CaCO_3$ classes even when they record the same proxy values (but could not, yet should be, distinguished in a traditional reactive-transport framework). Thus, IMP can be regarded as analogous to the multi-G model of Berner (1980), which separates bulk organic matter into multiple classes of organic compounds with different reactivities. However, the basis upon which we separate bulk $CaCO_3$ into multiple classes of $CaCO_3$ particles is more flexible, as these are not limited to reactivity but can be any combination of proxy signals as well as biological and physicochemical characteristics. In theory, IMP can simulate the effect of diagenesis and bioturbation on individual $CaCO_3$ particles by increasing the total number of $CaCO_3$ classes, although this results in an increased computational cost. Our new approach is the first combined diagenetic bioturbation model to pseudo-explicitly track proxy signals recorded in bulk $CaCO_3$ in the sediment column. This is realized by simulating the depth and time-dependent distribution of more than one $CaCO_3$ class each with distinct proxy signals.

In the following sections, we provide a detailed description of IMP in which the governing equations (Sect. 2.2), the numerical solutions (Sect. 2.3) and the simulation of signal tracking (Sect. 2.4) are highlighted. The default values of independent parameters (Table 1), the equations of dependent parameters (Table 2) and the equations of thermodynamic parameters (Table 3) are tabulated. The model code for IMP v.1.0 is available in Fortran90, MATLAB and Python (see Code availability).

### 2.2 Governing equations

For solid-phase species, IMP considers multiple ($n_{cc}$) classes of $CaCO_3$ particles, a single class of organic matter (OM) with the assumed chemical formula of $CH_2O$, and (a single class of) nonreactive detrital material (referred to as "clay" hereafter) to act as a "dilatant" and help determine the final burial velocity. The rate of change with time of the concentrations of these solid species in marine sediments are represented following the classic generalized equations of Boudreau (1997):

$$\frac{\partial (1-\phi)m_\theta}{\partial t} = -\frac{\partial (1-\phi)wm_\theta}{\partial z} - R_\theta$$
$$- (1-\phi)m_\theta \int_0^{z_{ml}} E_\theta(z,z')\,dz'$$
$$+ \int_0^{z_{ml}} \left\{1 - \phi(z')\right\} m_\theta(z') E_\theta(z',z)\,dz', \qquad (1)$$

where $m_\theta$ (mol cm$^{-3}$) represents the concentration of solid-phase species $\theta \in \{\ell, OM, clay;$ here $\ell = 1,2,\ldots,n_{cc}\}$; $\phi$ is the porosity; $t$ is the time (years); $E_\theta(z,z')$ represents the continuous exchange function (cm$^{-1}$ yr$^{-1}$), which describes transport of solid species $\theta$ from sediment depth $z$ (cm) to any other depth $z'$ (cm) (Sect. 2.2.2); $w$ is the burial velocity (cm yr$^{-1}$); $z_{ml}$ is the thickness of the mixed layer (cm); and $R_\theta$ (mol cm$^{-3}$ yr$^{-1}$) represents the net consumption rate of species $\theta$ through all biogeochemical reactions. On the right-hand side of Eq. (1), the total change in concentration of the solid species $\theta$ is expressed as the change due to advective transport (first term), biogeochemical reactions (second term) and bioturbational transport (third and fourth terms; note that there is no separate biodiffusion term).

For aqueous species, IMP considers dissolved oxygen ($O_2$), total dissolved $CO_2$ species (DIC) and carbonate alkalinity (ALK). The generalized equation for these aqueous species is given by Archer (1991):

$$\frac{\partial \phi c_\sigma}{\partial t} = \frac{\partial}{\partial z}\left(\frac{D_\sigma}{F}\frac{\partial c_\sigma}{\partial z}\right) + R_\sigma, \qquad (2)$$

where $c_\sigma$ represents the concentration (mol cm$^{-3}$), $D_\sigma$ is the diffusion coefficient (cm$^2$ yr$^{-1}$), $R_\sigma$ is the net production rate from all biogeochemical reactions (mol cm$^{-3}$ yr$^{-1}$) for aqueous species $\sigma \in \{O_2, DIC, ALK\}$ and $F$ represents the sediment formation factor (related to the tortuosity; Ullman and Aller, 1982).

**Table 1.** Values of independent parameters and boundary conditions.

| Parameter [units] | Symbol[a] | Value[b] | Ref./note[c] |
|---|---|---|---|
| **Independent parameters** | | | |
| Biodiffusion coefficient [$cm^2\,yr^{-1}$] | $D_{b,\theta}$ | 0.15 | 1 |
| Density of $CaCO_3$ [$g\,cm^{-3}$] | $\rho_\ell$ | 2.71 | 2 |
| Density of clay [$g\,cm^{-3}$] | $\rho_{clay}$ | 2.60 | 2 |
| Density of OM[d] [$g\,cm^{-3}$] | $\rho_{OM}$ | 1.2 | 3 |
| Homogeneous transport rate of sediment particles [$yr^{-1}$] | $P_{h,\theta}$ | 0.001 | 4 |
| Mixed-layer thickness [cm] | $z_{ml}$ | 12 | 5 |
| Molar mass of $CaCO_3$ [$g\,mol^{-1}$] | $M_\ell$ | 100 | 2 |
| Molar mass of clay [$g\,mol^{-1}$] | $M_{clay}$ | 258.16 | 2 |
| Molar mass of OM [$g\,mol^{-1}$] | $M_{OM}$ | 30 | 6 |
| Mole ratio of $O_2$ to OM consumed by oxic degradation of OM [dimensionless] | $\gamma_{O_2\text{-}OM}$ | 1.3 | 5 |
| Number of sediment grid points [dimensionless] | $N$ | 100 | 4 |
| OM/$CaCO_3$ rain ratio [dimensionless] | $r$ | 0.7 | 5 |
| Rate constant for $CaCO_3$ dissolution [$yr^{-1}$] | $k_{cc,\ell}$ | 365.25 | 5 |
| Rate constant for oxic degradation of organic matter [$yr^{-1}$] | $k_{oxic}$ | 0.06 | 1 |
| Rate constant for anoxic degradation of organic matter [$yr^{-1}$] | $k_{anoxic}$ | 0.06 | 1 |
| Reaction order for calcite dissolution [dimensionless] | $\eta_{cc}$ | 4.5 | 5 |
| **Boundary conditions** | | | |
| Carbonate alkalinity at seawater–sediment interface [mM] | | 2.285 | 4 |
| Oxygen concentration at seawater–sediment interface [mM] | | 0.165 | 4 |
| Salinity [‰] | $S$ | 35 | 4 |
| Temperature [°C] | $T_C$ | 2 | 7 |
| Total $CaCO_3$ rain flux [$\mu mol\,cm^{-2}\,yr^{-1}$] | $J_{CaCO_3}$ | 12 | 5 |
| Total concentration of aqueous $CO_2$ species at seawater–sediment interface [mM] | | 2.211 | 4 |
| Total sediment depth [cm] | $z_{tot}$ | 500 | 4 |
| Water depth [km] | $L$ | 3.5 | 8 |

[a] Given if defined in main text or used in equations in Tables 2 and 3. [b] Default values are given, which are used unless otherwise described. [c] (1) Emerson (1985). (2) From Robie and Hemingway (1995), assuming kaolinite ($Al_2Si_2O_5(OH)_4$) and calcite as representative clay and $CaCO_3$ phases, respectively. (3) A value close to the lower limit of the range (1.14–1.68 $g\,cm^{-3}$) reported by Mayer et al. (2004) is adopted (cf. Meyers, 2007). (4) Assumed. (5) Archer (1991). (6) Calculated assuming the chemical formula of OM as $CH_2O$. (7) Boudreau (1996). (8) Assumed, close to calcite saturation horizon and above calcite compensation depth in the modern oceans (e.g., Emerson and Archer, 1990; Oxburgh and Broecker, 1993). [d] OM denotes organic matter.

### 2.2.1 Biogeochemical reactions

Following Archer (1991), IMP considers degradation of organic matter and dissolution of $CaCO_3$ as the main biogeochemical reactions occurring in marine sediments. (In this version of IMP, we omit the role and geochemistry of opal and its dissolved pore-water phase, silicic acid; see however, e.g., Ridgwell et al. (2002), for a summary of the sedimentary system of opal).

The reaction term for organic matter is given by

$$R_{OM} = (1 - \phi)m_{OM}k_{OM}, \qquad (3)$$

where $k_{OM}$ is the first-order degradation rate constant for organic matter ($yr^{-1}$). To account for anaerobic degradation of organic matter by $SO_4$, IMP simulates an anoxic pathway below the dynamically calculated oxygen penetration depth ($z_{ox}$). Different rate constants for oxic ($k_{ox}$) and anoxic

($k_{anox}$) degradation can be adopted:

$$k_{OM} = \begin{cases} k_{ox} & (z \le z_{ox}) \\ k_{anox} & (z > z_{ox}) \end{cases}. \qquad (4)$$

Following Archer (1991), both rate constants are considered the same for the initial validation of our model in this study. While clearly an oversimplification, it serves as a first approximation of the importance of OM degradation on calcite dissolution and is also a requirement in order to be able to benchmark IMP to the model of Archer (1991). Although other pathways are used to degrade organic matter in marine sediments, such as nitrate and metal oxides, these have been shown to be quantitatively of less importance on a global scale (combined likely < 20 %; Archer et al., 2002; Thullner et al., 2009). It is, however, possible to artificially add DIC and ALK fluxes at a given depth, thereby simulating the production of ALK and DIC from a pathway that is not explicitly simulated (Supplement).

**Table 2.** Dependent parameters and their equations.

| Parameter [units] | Symbol[a] | Equation[b] | Ref./note[c] |
|---|---|---|---|
| Absolute temperature [K] | $T$ | $T = T_C + 273.15$ | |
| Concentration of aqueous $CO_2$ [mol cm$^{-3}$] | | $c_{ALK}/(K_1/[H^+] + 2K_1K_2/[H^+]^2)$ | 1 |
| Concentration of aqueous species $\sigma$ [mol cm$^{-3}$] | $c_\sigma$ | Eq. (2) | 2 |
| Concentration of bicarbonate ion [mol cm$^{-3}$] | | $c_{ALK}/(1 + 2K_2/[H^+])$ | 1 |
| Concentration of carbonate ion [mol cm$^{-3}$] | $c_{CO_3^{2-}}$ | $c_{CO_3^{2-}} = c_{ALK}/([H^+]/K_2 + 2)$ | 1 |
| Concentration of $H^+$ [mol kg$^{-1}$] | $[H^+]$ | $[H^+] = [-K_1(1 - c_{DIC}/c_{ALK}) + \{K_1^2(1 - c_{DIC}/c_{ALK})^2 \\ -4K_1K_2(1 - 2c_{DIC}/c_{ALK})\}^{0.5}]/2$ | 1 |
| Concentration of solid species $\theta$ [mol cm$^{-3}$] | $m_\theta$ | Eq. (1) | 2 |
| Detrital rain flux [μg cm$^{-2}$ yr$^{-1}$] | | $(1/9)J_{CaCO_3}M_\ell$ | 1 |
| Diffusion coefficient for ALK [cm$^2$ yr$^{-1}$] | $D_{ALK}$ | $D_{ALK} = 151.69 + 7.93T_C$ | 3 |
| Diffusion coefficient for DIC [cm$^2$ yr$^{-1}$] | $D_{DIC}$ | $D_{DIC} = 151.69 + 7.93T_C$ | 3 |
| Diffusion coefficient for dissolved $O_2$ [cm$^2$ yr$^{-1}$] | $D_{O_2}$ | $D_{O_2} = 348.62 + 14.09T_C$ | 3 |
| Formation factor [dimensionless] | $F$ | $F = \phi^{-3}$ | 4 |
| Molar volume [cm$^3$ mol$^{-1}$] | $V_\theta$ | $V_\theta = M_\theta/\rho_\theta$ | 2 |
| OM rain flux [μmol cm$^{-2}$ yr$^{-1}$] | | $rJ_{CaCO_3}$ | 1 |
| Porosity [dimensionless] | $\phi$ | $\phi = 0.1932\exp(-z/3) + 0.8068$ | 5 |
| Pressure [bar] | $p$ | $p = 100L$ | 6 |
| Saturation degree of calcite [dimensionless] | $\Omega_{cc}$ | $\Omega_{cc} = c_{CO_3^{2-}} \times 10^{-3}$ TS1 $\times 10.3 \times 10^{-3}/K_{cc}$ | 1,7 |
| Sediment depth [cm] | $z$ | $z = z_{tot} \times \ln\{(\beta + \zeta^2)/(\beta - \zeta^2)\}/\ln\{(\beta + 1)/(\beta - 1)\}$ | 8 |

[a] Given if defined in main text or used in equations in Tables 2 and 3. [b] Parameter values are calculated based on the listed equations unless otherwise described. [c] (1) Archer (1991). (2) Sect. 2. (3) Hülse et al. (2018). (4) Ullman and Aller (1982). (5) Archer (1996). No porosity dependence on $CaCO_3$ is assumed. (6) Approximate relation, cf., Saunders and Fofonoff (1976). (7) Dissolved calcium concentration is assumed to be constant at 10.3 mM. (8) Modified after Eq. (9-32) of Hoffman and Chiang (2000, chap. 9), where $\zeta$ denotes the normalized regular grid and $\beta = 5 \times 10^{-11} + 1$.

The reaction term for any class $\ell$ of $CaCO_3$ particles is given by

$$R_\ell = (1 - \phi)m_\ell k_{cc,\ell}(1 - \Omega_{cc})^{\eta_{cc}} H(1 - \Omega_{cc}), \quad (5)$$

where $k_{cc,\ell}$ is the rate constant (yr$^{-1}$); $\Omega_{cc}$ is the saturation degree; $\eta_{cc}$ is the reaction order for $CaCO_3$ dissolution; and the Heaviside function $H$ guarantees that net $CaCO_3$ precipitation does not occur (Archer, 1991). Note that the model allows assignment of different dissolution rate constants to different classes of $CaCO_3$ particles (e.g., Keir, 1980). For this study, however, unless otherwise described, we assume a dissolution rate of $k_{cc,\ell} = 365.25$ yr$^{-1}$ for all classes, which is a value determined by Archer (1991). CE1

The clay species is assumed to be nonreactive. Hence,

$$R_{clay} = 0. \quad (6)$$

The reaction terms for aqueous species $O_2$, DIC and ALK are correspondingly given by (Archer, 1991)

$$R_{O_2} = -\gamma_{O_2\text{-}OM}(1 - \phi)m_{OM}k_{ox}, \quad (7)$$

$$R_{DIC} = R_{OM} + \sum_{\ell=1}^{n_{cc}} R_\ell, \quad (8)$$

$$R_{ALK} = (1 - \phi)m_{OM}k_{anox} + 2\sum_{\ell=1}^{n_{cc}} R_\ell. \quad (9)$$

Here, $\gamma_{O_2\text{-}OM}$ in Eq. (7) is the mole ratio of oxygen to organic matter consumed upon oxic degradation of organic matter. We assume that the aqueous carbonate system is always at equilibrium, and we calculate the partitioning of the aqueous carbonate species ($H_2CO_3$, $HCO_3^-$ and $CO_3^{2-}$) based on alkalinity and DIC concentrations in conjunction with the apparent equilibrium dissociation constants adjusted for pressure, salinity and temperature (Tables 2, 3). Other options to

**Table 3.** Thermodynamic parameters.

| Parameter [units] | Symbol[a] | Equation | Ref./note[b] |
|---|---|---|---|
| Equilibrium constant for carbonic acid dissociation [$mol\,kg^{-1}$] | $K_1$ | $-\log K_1 = -126.34048 + 6320.813/T + 19.568224 \times \ln T$ $+13.4191 \times S^{0.5} + 0.0331 \times S - 5.33 \times 10^{-5} \times S^2$ $+(-530.1228 \times S^{0.5} - 6.103 \times S)/T - 2.06950 \times S^{0.5} \times \ln T$ $-\{-(-25.50 + 0.1271 \times T_C) \times p + 0.5 \times (-3.08 \times 10^{-3}$ $+0.0877 \times 10^{-3} \times T_C) \times p^2\}/83.131/T/\ln 10$ | 1 |
| Equilibrium constant for bicarbonate dissociation [$mol\,kg^{-1}$] | $K_2$ | $-\log K_2 = -90.18333 + 5143.692/T + 14.613358 \times \ln T$ $+21.0894 \times S^{0.5} + 0.1248 \times S - 0.0003687 \times S^2$ $+(-772.483 \times S^{0.5} - 20.051 \times S)/T - 3.32254 \times S^{0.5} \times \ln T$ $-\{-(-15.82 - 0.0219 \times T_C) \times p + 0.5 \times (1.13 \times 10^{-3}$ $-0.1475 \times 10^{-3} \times T_C) \times p^2\}/83.131/T/\ln 10$ | 1 |
| Solubility product of calcite [$mol^2\,kg^{-2}$] | $K_{cc}$ | $-\log K_{cc} = -171.9065 - 0.077993 \times T + 2839.319/T + 71.595 \times \log T$ $+(-0.77712 + 0.0028426 \times T + 178.34/T) \times S^{0.5} - 0.07711 \times S$ $+0.0041249 \times S^{1.5} - \{-(-48.76 + 0.5304 \times T_C) \times p$ $+0.5 \times (-11.76 \times 10^{-3} + 0.3692 \times 10^{-3} \times T_C) \times p^2\}/83.131/T/\ln 10$ | 2 |

[a] Given if defined in main text or used in equations in Tables 2 and 3. [b] (1) Millero (1995); Millero et al. (2006). (2) Mucci (1983); Millero (1995)

utilize published routines for the calculation of the aqueous carbonate system, mocsy 2.0 (Orr and Epitalon, 2015) and CO2SYS (Lewis and Wallace, 1998; van Heuven et al., 1998; Humphreys et al., 2020), are presented in the Supplement.

### 2.2.2 Bioturbation

Bio-mixing of solid-phase species in the model is simulated by means of a transition matrix. A wide range of bio-mixing styles can be captured by the transition matrix because a transport probability of solid particles from one sediment layer to another can be specified with the value of a cell whose row and column numbers correspond to the two layers between which particles are transported. Thus, the use of the transition matrix facilitates the implementation of user-defined/biology-based particle mixing, whether local or non-local (e.g., Trauth, 1998; Shull, 2001). In this section, we elaborate upon how the bioturbation term in Eq. (1) can be derived from the transition matrix.

The rate at which particles of solid species $\theta$ are transported from layer $i$ to layer $j$, $P_{\theta,ij}$ ($yr^{-1}$), is given by

$$P_{\theta,ij} = \frac{N_{\theta,ij}}{\sum_{j=1}^{n_{ml}} N_{\theta,ij}} \frac{1}{\tau}, \tag{10}$$

where $N_{\theta,ij}$ is the number of particles of species $\theta$ moved from layer $i$ to layer $j$, $n_{ml}$ is the total number of layers within the bioturbated zone and $\tau$ is the time (years) required for the displacements. Note that $P_{\theta,ij} \times \tau$ represents the particle transport probability and corresponds to components at $(i, j)$ of the transition matrix (Trauth, 1998; Shull, 2001). When bioturbation causes mixing of sediment particles based on the above transport rate, the number of particles of species $\theta$ in layer $i$ changes with time according to the following equation:

$$\frac{dN_{\theta,i}}{dt} = -N_{\theta,i} \sum_{j=1}^{n_{ml}} P_{\theta,ij} + \sum_{j=1}^{n_{ml}} N_{\theta,j} P_{\theta,ji}, \tag{11}$$

where $N_{\theta,i}$ is the total number of particles of species $\theta$ in layer $i$ (compare Eq. 11 with Eq. 3.117 of Boudreau, 1997).

The concentration of species $\theta$ in layer $i$, $m_{\theta,i}$ ($mol\,cm^{-3}$), can be given by (cf., Boudreau, 1997)

$$(1 - \phi_i)m_{\theta,i} \equiv \frac{\alpha_\theta N_{\theta,i}}{A \delta z_i}, \tag{12}$$

where $\phi_i$ and $\delta z_i$ are the porosity and the thickness (cm) of layer $i$, respectively; $\alpha_\theta$ represents the moles of species $\theta$ (mol) included in one particle; and $A$ is the cross-sectional area in the model ($cm^2$). One can then deduce the following from Eqs. (11) and (12):

$$\frac{d(1 - \phi_i)m_{\theta,i}}{dt} = -(1 - \phi_i)m_{\theta,i} \sum_{j=1}^{n_{ml}} P_{\theta,ij}$$
$$+ \sum_{j=1}^{n_{ml}} (1 - \phi_j)\frac{\delta z_j}{\delta z_i} m_{\theta,j} P_{\theta,ji} \tag{13}$$

(compare Eq. 13 with Eq. 3.118 of Boudreau, 1997). Equation (13) can be simplified with a modified transition matrix for species $\theta$, with components at $(i, j)$ denoted as $K_{\theta,ij}$ and calculated based on the particle transport rate $P_{\theta,ij}$:

$$K_{\theta,ij} = \begin{cases} \delta z_i P_{\theta,ij}/\delta z_j & (i \neq j) \\ -\sum_{j\neq i}^{n_{ml}} P_{\theta,ij} & (i = j). \end{cases} \tag{14}$$

Using Eq. (14), we can rewrite Eq. (13) as a function of $K_{\theta,ij}$:

$$\frac{d(1 - \phi_i)m_{\theta,i}}{dt} = \sum_{j}^{n_{ml}} (1 - \phi_j)m_{\theta,j} K_{\theta,ji}. \tag{15}$$

Formulation of bioturbation in a continuum system needs a corresponding continuous function. We define a continuous exchange function $E_\theta$ (cm$^{-1}$ yr$^{-1}$) as follows (cf., Boudreau, 1997):

$$E_\theta(z_i, z_j) \equiv \lim_{\delta z_j \to 0} (P_{\theta,ij}/\delta z_j), \tag{16}$$

where $z_i$ and $z_j$ denote the depths of sediment layer $i$ and $j$, respectively. With Eq. (16), we can write a continuous form of Eq. (13) in the limits of zero thicknesses for discretized sediment layers:

$$\frac{\partial (1-\phi)m_\theta}{\partial t} = -(1-\phi)m_\theta \int_0^{z_{ml}} E_\theta(z, z')\, dz'$$
$$+ \int_0^{z_{ml}} \{1 - \phi(z')\}m_\theta(z') E_\theta(z', z)\, dz'. \tag{17}$$

Here, $z'$ denotes any depth except at $z$, and $z_{ml}$ is the thickness of the mixed layer. Eq. (17) is the same as Eq. (3.121) of Boudreau (1997) and the two bioturbation terms in Eq. (1). Note that Eq. (15) is a finite difference version of Eq. (17), and the transition matrix corrected for porosity therein (i.e., $(1-\phi_i)K_{\theta,ij}$ representing components at $(i, j)$) corresponds to the bioturbational transport part of the Jacobian matrix for species $\theta$, which is used for solving the governing equations (Sect. 2.3).

Three different transition matrices were created for the present study to illustrate different styles of bio-mixing (Fig. 1): Fickian mixing, homogeneous mixing and the more mechanistic automaton-based mixing simulated by the particle-tracking bioturbation simulator LABS (e.g., Boudreau et al., 2001; Choi et al., 2002; Kanzaki et al., 2019).

The transition matrix that assumes Fickian diffusion for bioturbation (parameterized with $D_{b,\theta}$, Goldberg and Koide, 1962), can be expressed by

$$K_{\theta,ij} = \begin{cases} -K_{\theta,ij}(j=i+1) & (i=j=1) \\ -K_{\theta,ij}(j=i+1) & \\ \quad -K_{\theta,ij}(j=i-1) & (1 < i = j < n_{ml}) \\ -K_{\theta,ij}(j=i-1) & (i=j=n_{ml}) \\ \{(1-\phi_i)D_{b,\theta,i} & \\ \quad +(1-\phi_j)D_{b,\theta,j}\}/ & \\ \{\delta z_i(1-\phi_i) & \\ \quad \cdot(\delta z_i + \delta z_j)\} & (2 \le j = i+1 \le n_{ml} \text{ or} \\ & 1 \le j = i-1 \le n_{ml}-1) \\ 0 & (\text{else}), \end{cases} \tag{18}$$

where $D_{b,\theta,i}$ represents the biodiffusion coefficient for solid species $\theta$ at sediment layer $i$. As a default biodiffusion parameterization, a depth-independent value of 0.15 cm$^2$ yr$^{-1}$ is assumed (Emerson, 1985, Table 1). Note that the biodiffusion considered in this study is only intraphase biodiffusion and does not include interphase biodiffusion (e.g., Meysman

et al., 2005; Munhoven, 2021). The implementation of interphase biodiffusion requires a different transition matrix.

The transition matrix for homogeneous mixing can be given by

$$K_{\theta,ij} = \begin{cases} \delta z_i P_{h,\theta}/\delta z_j & (i \ne j \text{ and } 1 \le i, j \le n_{ml}) \\ -(n_{ml}-1)P_{h,\theta} & (1 \le i = j \le n_{ml}) \\ 0 & (\text{else}), \end{cases} \tag{19}$$

where $P_{h,\theta}$ (yr$^{-1}$) is the homogeneous transport rate for solid species $\theta$. A value of $10^{-3}$ yr$^{-1}$ is assumed for the default homogeneous mixing (Table 1).

To obtain the mechanistic automaton-based transition matrix, we utilized the eLABS v.0.2 code, the latest release of lattice-automaton bioturbation simulator (LABS) by Kanzaki et al. (2019), with which a transition matrix can be extracted based on Eqs. (10) and (14). The new features of LABS added by Kanzaki et al. (2019), i.e., 2-D porewater flow and diagenesis, were disabled to extract mixing controlled dominantly by benthos biology as in Boudreau et al. (2001). A 200-year LABS simulation was run with a deposit feeder with a body size of $0.25 \times 0.25 \times 1.65$ cm$^3$, a locomotion speed of 10 cm d$^{-1}$ and a maximum ingestion rate of 1 g of sediment per gram of organism per day in a $0.25 \times 12 \times 15$ cm$^3$ 3-D sediment system. Transition matrices were created every 10 model days (cf. Reed et al., 2007), and the averaged transition matrix over 200 model years multiplied by a factor of 1/10 was adopted to represent the transition matrix derived from the above LABS simulation. The factor of 1/10 was introduced above because the LABS mixing would otherwise have a relatively high mixing intensity (equivalent to a biodiffusion coefficient of 0.1–10 cm$^2$ yr$^{-1}$; cf. Kanzaki et al., 2019) and also to facilitate the numerical solution of the model with the LABS mixing (see below).

The default transition matrices, corrected for porosity, are shown in Fig. 1. Fickian mixing is a local mixing, allowing translocation of particles only between adjacent sediment layers, resulting in a tridiagonal matrix (Fig. 1a). On the other hand, nonlocal mixing (homogeneous and LABS mixing) allows the transportation of particles between remote layers and, thus, is characterized with the spread of nonzero components away from the main diagonal in the transition matrix (Fig. 1b, c). As defined in Eq. (19), the transition matrix for homogeneous mixing has components that systematically change with rows and columns (Fig. 1b) compared with the transition matrix for LABS mixing that has more randomly spread noncontinuous values (Fig. 1c). The porosity-corrected transition matrix corresponds to the bioturbational transport part of the Jacobian matrix used for solving the governing equations (Sect. 2.3). Therefore, the difficulty to achieve a numerical solution of the model differs between chosen mixing styles reflecting corresponding transition matrices: in general, this is the least difficult with Fickian mixing and the most difficult with LABS mixing (Fig. 1).

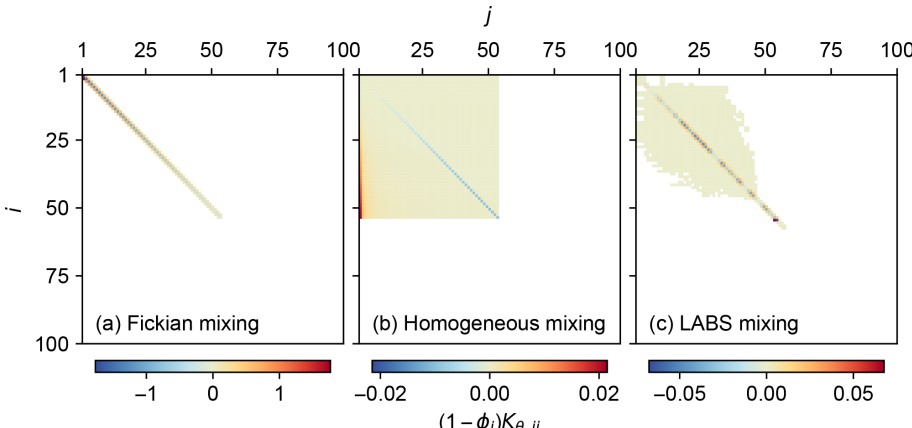

**Figure 1.** Transition matrices corrected for porosity, representing three different bio-mixing styles: **(a)** Fickian intraphase biodiffusion, **(b)** homogeneous mixing and **(c)** automaton-based mixing by a particle-tracking bioturbation simulator LABS.

Note that the transition matrices for Fickian and homogeneous mixing change with assumed mixed-layer depth (related to $n_{ml}$) and/or parameters that define mixing intensity ($D_{b,\theta}$ and $P_{h,\theta}$) as in Eqs. (18) and (19) (cf. Table 1). Additional LABS simulations, with variations related to deposit-feeder behavior and/or modified sediment grid dimensions, are necessary to generate a new LABS-based transition matrix.

### 2.2.3 Burial velocity/advection

The burial velocity in IMP changes according to the volume change of solid material caused by biogeochemical reactions and bio-mixing because a constant, time-independent porosity profile is assumed (Eq. 23). This section describes how the change in burial rate is calculated in the model.

Multiplying the governing equation (Eq. 1) by the molar volume $V_\theta$ (cm$^3$ mol$^{-1}$) for solid species $\theta$ leads to

$$\frac{\partial (1-\phi)V_\theta m_\theta}{\partial t} = -\frac{\partial (1-\phi)w V_\theta m_\theta}{\partial z} - V_\theta R_\theta$$

$$+ V_\theta \left[ -(1-\phi)m_\theta \int_0^{z_{ml}} E_\theta(z,z')\, dz' \right.$$

$$\left. + \int_0^{z_{ml}} \{1-\phi(z')\}m_\theta(z')E_\theta(z',z)\, dz' \right]. \quad (20)$$

Note that the molar volume $V_\theta$ can be obtained from the density, $\rho_\theta$ (g cm$^{-3}$), and the molar mass, $M_\theta$ (g mol$^{-1}$), of species $\theta$ as $V_\theta = M_\theta/\rho_\theta$. Summing Eq. (20) for all solid-phase species gives

$$\frac{\partial (1-\phi)w}{\partial z} = -\sum_\theta V_\theta R_\theta$$

$$+ \sum_\theta V_\theta \left[ -(1-\phi)m_\theta \int_0^{z_{ml}} E_\theta(z,z')\, dz' \right.$$

$$\left. + \int_0^{z_{ml}} \{1-\phi(z')\}m_\theta(z')E_\theta(z',z)\, dz' \right]. \quad (21)$$

For the derivation of Eq. (21), the following relations are enforced:

$$\sum_\theta V_\theta m_\theta = 1, \quad (22)$$

$$\frac{\partial \phi}{\partial t} = 0. \quad (23)$$

Equations (22) and (23) express the constraint that the volume fractions of all solid species sum to 1 cm$^3$ cm$^{-3}$ and the assumption of time independency of porosity, respectively. Unless bio-mixing is Fickian (intraphase) biodiffusion with the same intensity and the same mixed-layer depth for all solid species (see below), the burial velocity is calculated based on Eq. (21).

If bio-mixing of solid species $\theta$ is Fickian biodiffusion with a coefficient $D_{b,\theta}$ (cm$^2$ yr$^{-1}$), Eq. (20) can be expressed as

$$\frac{\partial (1-\phi)V_\theta m_\theta}{\partial t} = -\frac{\partial (1-\phi)w V_\theta m_\theta}{\partial z} - V_\theta R_\theta$$

$$+ \frac{\partial}{\partial z}\left\{ (1-\phi)D_{b,\theta}\frac{\partial V_\theta m_\theta}{\partial z} \right\}. \quad (24)$$

Further, if bio-mixing of all solid species occurs as Fickian biodiffusion with the same mixing intensity ($D_b$) and depth

$(z_{ml})$, Eqs. (22) and (23) lead to a simpler burial velocity equation:

$$\frac{\partial (1-\phi)w}{\partial z} = -\sum_\theta V_\theta R_\theta. \qquad (25)$$

Therefore, when the transition matrix is specified to represent intraphase biodiffusion (e.g., Fig. 1a) and the same matrix is applied to all solid species, Eq. (25) is used to calculate burial velocity, otherwise Eq. (21) is used. In either case, the model generally satisfies Eq. (22).

## 2.3 Initial conditions, boundary conditions and numerical solutions

### 2.3.1 Initial and boundary conditions

At the beginning of the calculation, we must define both initial (e.g., solid and pore-water composition) and boundary conditions as well as the structure of the grid.

In the default setting of IMP, the calculation domain represents a $z_{tot} = 500\,\text{cm}$ sediment column and is discretized into $N = 100$ layers whose thickness increases with depth from less than $10^{-2}$ to more than $10^2$ cm following a logarithmic function (Table 2). Furthermore, a time-independent exponential porosity profile is imposed (Table 2). One may modify the grid structure and porosity profile by changing the associated parameter values (Table 2) defined in the code (Supplement).

As initial conditions for the sediment grid, the model assumes almost nonexistent concentrations of $10^{-8}\,\text{mol cm}^{-3}$ for all solid species (carbonate, organic matter and clay), and the volume deficiency relative to the solid space prescribed by the assumed porosity is filled by the later time-integration (see below). Ambient ocean concentrations at the seawater–sediment interface are adopted as the initial concentrations for all aqueous species at all depths. These initial values, however, do not have an impact on our results, as the model is run to steady state before an experiment is started (e.g., a proxy signal change event is simulated).

The upper boundary conditions at the seawater–sediment interface are given by mass fluxes of simulated solid species and concentrations for simulated aqueous species (Tables 1, 2). The lower boundary conditions at $z_{tot}$ for all aqueous species are given by zero concentration-gradients. If oxygen is consumed within the simulated sediment column (i.e., $z_{ox} < z_{tot}$), the dynamically calculated oxygen penetration depth marks a lower boundary for oxygen (i.e., $c_{O_2} = 0$ at $z = z_{ox}$). As boundary conditions can change with model time (e.g., in the proxy signal change experiments), they are specified before each time integration.

### 2.3.2 Program structure and numerical solution

Solutions for the temporal and spatial evolution of individual solid and aqueous species are obtained by solving the governing equations with the finite difference method (e.g., Hoffman and Chiang, 2000). Figure 2 summarizes the structure of the code to solve the governing equations, and the calculation at a given time is conducted by the model in the following four main steps.

1. First, organic matter and oxygen concentration profiles are calculated using Eqs. (1) and (2) (for $\theta = $ "OM" and $\sigma = $ "$O_2$"). As both calculations depend on the oxygen penetration depth $z_{ox}$, they are conducted iteratively by the following steps (cf. Emerson, 1985; Archer, 1991):

   a. $z_{ox}$ is calculated based on the $O_2$ profile from the previous iteration or time instance;

   b. the OM profile is updated based on the $z_{ox}$ from step a;

   c. $(N+1)$ cases of the $O_2$ profile are calculated, each of which assumes that $z_{ox}$ is located in one of $N$ sediment layers or below the model sediment domain with the corresponding boundary conditions (Sect. 2.3.1) using the aerobic degradation rates calculated from the OM profile obtained in step b;

   d. among the $(N+1)$ cases of step c, the $O_2$ profile that is most consistent with the boundary conditions (i.e., $c_{O_2} = 0$ at $z = z_{ox} < z_{tot}$ or $c_{O_2} > 0$ at $z = z_{tot} < z_{ox}$) is adopted with the corresponding $z_{ox}$; and

   e. steps a–d are repeated until $z_{ox}$ in steps a and d are located in the same sediment layer or both below the model sediment domain. After the convergence of the above iteration, anoxic degradation of OM is calculated at $z > z_{ox}$ if $z_{ox} < z_{tot}$.

2. Second, with the obtained oxic and anoxic decomposition of organic matter, concentration profiles of multiple classes of $CaCO_3$, DIC and ALK are solved (Eqs. 1 and 2 for $\theta = \ell$ and $\sigma = $ "DIC" and "ALK") in a fully coupled way (e.g., Steefel and Lasaga, 1994, see below). Concentrations of individual aqueous carbonate species and pH are calculated based on the obtained ALK and DIC profiles assuming charge balance and equilibria for dissociations of carbonic acid and bicarbonate ion (Tables 2, 3; Archer, 1991).

3. Third, the clay concentration is calculated using Eq. (22) and the concentrations of OM and $n_{cc}$ classes of $CaCO_3$ obtained in steps 1 and 2, following Munhoven (2021). Obtained clay concentration is substituted into Eq. (1) for $\theta = $ "clay" to confirm the satisfaction of the governing equation.

4. Lastly, the reaction and bioturbation terms for solid species are used to update burial velocity using either Eq. (21) or (25). When the updated burial velocity is significantly different from the previous velocity, iteration is conducted (i.e., calculations of all species are

conducted again with the updated burial velocity) until the relative difference becomes negligible ($\leq 10^{-6}$) within the same time step (Fig. 2). If the above criterion is not met within 20 iterations (only encountered in a few conditions in lysocline experiments; Sect. 3.1), the results yielding the minimum relative difference are adopted (still less than a few percent). The procedures in steps 3 and 4 ensure that the volume fractions of solid species sum to 1 cm$^3$ cm$^{-3}$ (Eq. 22).

The concentration profiles of individual species are solved based on the difference equations of Eqs. (1) and (2), which are obtained by the finite difference method. The second-order and first-order spatial differential terms are discretized by the second-order central and the first-order upwind differencing schemes, respectively (e.g., Hoffman and Chiang, 2000). The finite difference form of the bioturbation term in Eq. (1) is formulated with a transition matrix (Eq. 15). The difference equations for all the solid and aqueous species are solved time-implicitly (e.g., Steefel and Lasaga, 1994). For the solution of the difference equations that are nonlinear, as is the case for the carbonate system (multiple CaCO$_3$ classes, DIC and ALK), the Newton–Raphson method is utilized (Fig. 2) where the solution is iteratively updated along with the Jacobian matrix until its relative difference from the previous iteration becomes $\leq 10^{-6}$ (e.g., Steefel and Lasaga, 1994). Note that the porosity-corrected transition matrix corresponds to the bioturbational transport part of the Jacobian matrix (Fig. 1).

The time step taken for the time integration of the governing equations can vary between and within simulations, and can be specified by the user (cf. Sect. 3.1). In the default setting, the time step increases with model time from 100 to $10^5$ years to reach steady state (e.g., a spin-up phase of simulation prior to imposing a signal change event; Sects. 2.4, 3), and a smaller and fixed time step is taken when simulating a signal change event (5 or 10 years for a 10 or 50 kyr signal change event, respectively) as well as its aftermath (Sects. 2.4, 3).

By default, the model monitors and records depth-integrated fluxes of individual rate terms in Eq. (1) or (2) (fluxes caused by amount change in sediment, sediment rain, biogeochemical reactions, advection, bio-mixing and so on) for each solid/aqueous species, as well as the residual flux as a sum of all the fluxes, which is ideally zero, to confirm the mass balance of the species. The residual fluxes for all the solid and aqueous species are negligible (e.g., $\leq 10^{-6}$ times the rain fluxes) for all the simulations presented in this paper (Sect. 3).

## 2.4  Signal tracking

### 2.4.1  Tracking input signals

Tracking of proxy signals in carbonates is conducted by assigning different numerical values to the simulated CaCO$_3$ classes and by scaling their input fluxes to reflect the overall change in proxy signal with time. Thus, proxy signal changes are reflected as changes in the boundary conditions (i.e., rain fluxes of different CaCO$_3$ classes) in the model (see Sect. 2.3). Assignment of proxy signals and fluxes to CaCO$_3$ classes can be realized by three methods (Fig. 3).

In the first method (a "time-stepping" method), any change in proxy signal is approximated by a step function, i.e., a continuously varying analogue signal is (digitally) discretized (see, e.g., dotted curve in the top panel of Fig. 3). Each step is represented by a separate and unique CaCO$_3$ class, characterized by the approximate proxy value (Fig. 3a). For example, if a signal change event is discretized into 10 steps, 10 different CaCO$_3$ classes with unique proxy values are simulated. Any change in a proxy signal during each discretized time interval is thus muted. Accordingly, the accuracy of the proxy signal approximation is increased by increasing the number of steps and thus the number of simulated CaCO$_3$ classes, which, however, results in an increased computation cost (Supplement). As an advantage, one can track any number of proxies as long as the signal changes of all tracked proxies occur within a simulated event (Supplement).

The second method to assign proxy signals (an interpolating method) simulates only the end-member CaCO$_3$ classes, each of which possesses a unique combination of the maximum and/or minimum input-signal values. As an example, one proxy can be tracked with two CaCO$_3$ classes, with the first possessing the maximum and the second the minimum proxy value. Intermediate values of an input proxy are realized by assigning varying fluxes to the two end-member classes such that the sum of their flux-weighted values results in the input-signal value at each time step (Fig. 3b). Accordingly, the input proxy signal is always accurately represented regardless of the resolution of the time discretization. As a disadvantage of method 2, the number of simulated CaCO$_3$ classes increases with the number of proxies to be tracked. In general, $2^{n_p}$ classes of CaCO$_3$ particles are necessary when tracking $n_p$ proxies because the number of unique combinations of the maximum and/or minimum signal values is increased by a factor of 2 for every additional proxy to be tracked:

$$\underbrace{\overset{\text{proxy 1}}{2} \times \overset{\text{proxy 2}}{2} \times \cdots \times \overset{\text{proxy } n_p}{2}}_{n_p} = 2^{n_p}. \tag{26}$$

Nonetheless, the computational demand is lower compared with method 1 in most cases because we are interested in a limited number of proxies and, thus, fewer CaCO$_3$ classes are simulated ($2^{n_p}$ in method 2 < time steps in method 1).

The third method (a direct tracking method) separates bulk CaCO$_3$ into multiple classes based on how the simulated proxies are determined. For example, when the tracked proxy is $\delta^{13}$C, which is determined by the $^{13}$C/$^{12}$C ratio (X in Fig. 3), method 3 simulates classes of Ca$^{13}$CO$_3$ and Ca$^{12}$CO$_3$ (Y and G, respectively; Fig. 3c). The rain fluxes

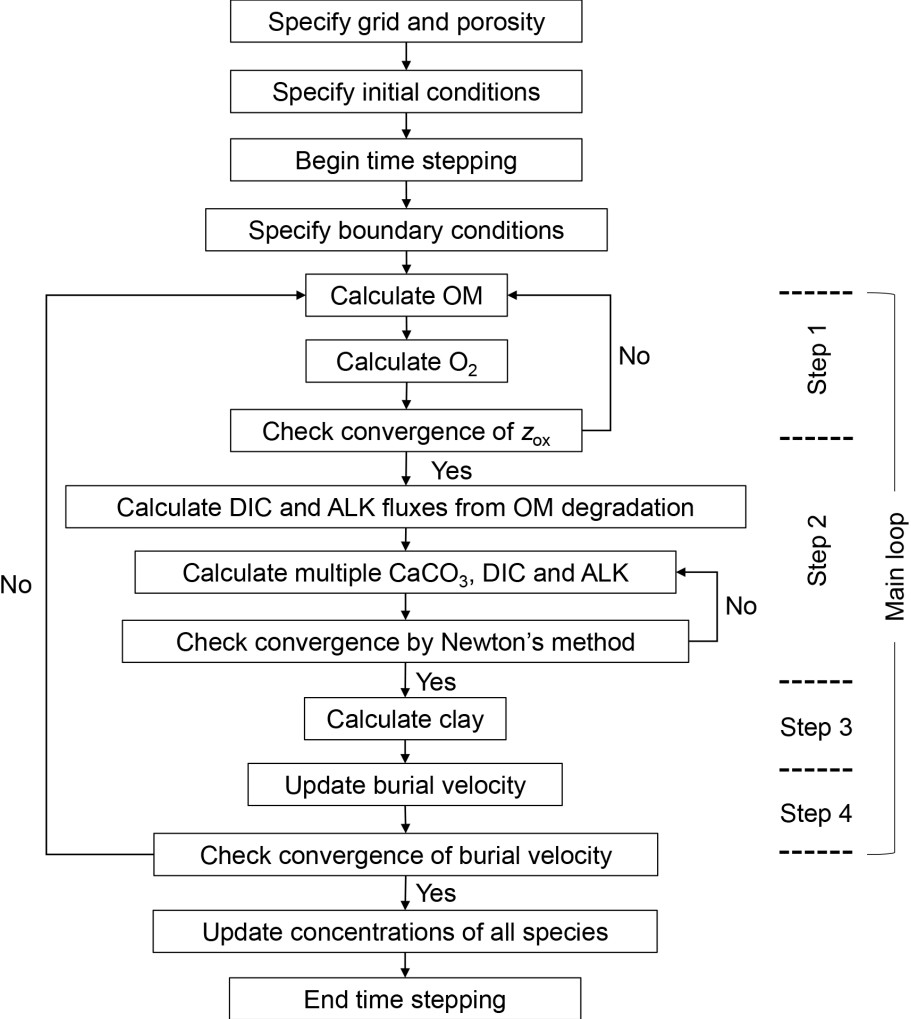

**Figure 2.** Program structure for reactive-transport modeling of diagenesis.

of individual classes at a given time step are directly calculated based on the definition of the proxy and the contemporaneous proxy value (see boxes in Fig. 3c). Thus, one can regard method 3 as a derivative of method 2 that defines the end-member $CaCO_3$ classes based on the definition of the tracked proxy. Because the flux calculation must change with the simulated proxy signal, method 3 is not as flexible as methods 1 and 2, but the computational effort can be further reduced because a certain class can be used to define multiple proxies (e.g., $Ca^{12}CO_3$ is related to the definition of both $^{14}C$ age and $\delta^{13}C$). Method 3 has the unique advantage of enabling additional biogeochemical reaction terms for any specific $CaCO_3$ class if necessary. For instance, when tracking $^{14}C$ age, one needs to account for the radioactive decay of $Ca^{14}CO_3$ and the accompanied generation of alkalinity, which can be implemented with method 3. Currently method 3 tracks four proxies including $^{14}C$ age with five $CaCO_3$ classes (Supplement).

After the signal and flux assignment by any of the three methods, the model is spun up to steady state with only the $CaCO_3$ class(es) with pre-event proxy values being deposited to sediment (Fig. 3). After the spin-up, a proxy signal change event is simulated by changing the rain fluxes of different $CaCO_3$ classes with different proxy values (i.e., the boundary conditions) with model time (Fig. 3). After the signal change event, the model is run until a new steady state is reached.

Note that the methods and procedures described above can be applied not only to track proxy signals but also any other property of $CaCO_3$ particles such as particle size and deposition time (cf. Sect. 2.4.2). In this case, methods 2 and 3 track the property in the same way (Sect. 2.4.2; Supplement).

### 2.4.2 Tracking signals within the sediment

After input signals are reflected in rain fluxes by any of the three methods in Sect. 2.4.1, they are modified within the sediment by bioturbation and chemical erosion. Caution

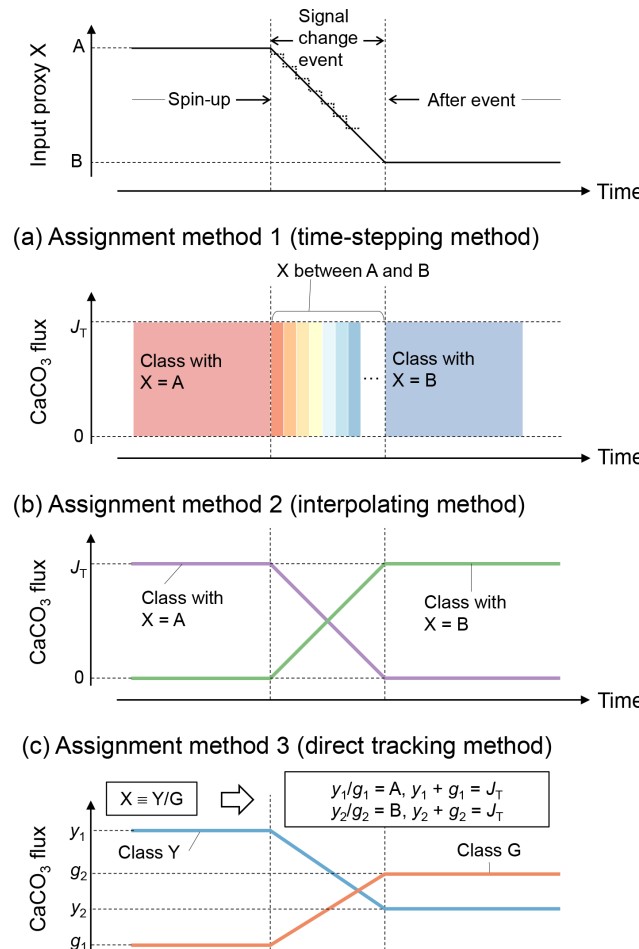

**Figure 3.** Schematic of signal tracking simulation. Input proxy signal X (solid line in the uppermost panel) is reflected in rain fluxes of multiple classes of $CaCO_3$ particles using three different methods **(a–c)**. Method 1 **(a)** approximates the input proxy signal by a step function (dotted line in the uppermost panel) and uses different classes of $CaCO_3$ with separate and unique proxy values at individual time steps. The rain flux of each $CaCO_3$ class can take either 0 or the total rain flux value $J_T$. Method 2 **(b)** uses $CaCO_3$ classes with the maximum and minimum values of proxy (A and B), and rain fluxes of these $CaCO_3$ classes are changed so that flux-weighted sums of proxy values of $CaCO_3$ classes become the same as the input proxy values. Method 3 **(c)** separates bulk $CaCO_3$ into $CaCO_3$ classes that define the proxy signal (classes Y and G), and rain fluxes of these $CaCO_3$ classes are calculated based on the proxy signal values (see boxes). See Sect. 2.4.1 for more details.

needs to be taken with respect to numerical diffusion, which is inevitably introduced to the difference form of the advection term (first term on the right-hand side of Eq. 1) in a finite difference approach (e.g., Hoffman and Chiang, 2000; Steiner et al., 2016). For an accumulating column of sediment in a fixed grid, numerical diffusion artificially mixes the deposited and buried sediment particles along with their proxy signals, especially at depths where grid cells are rel-

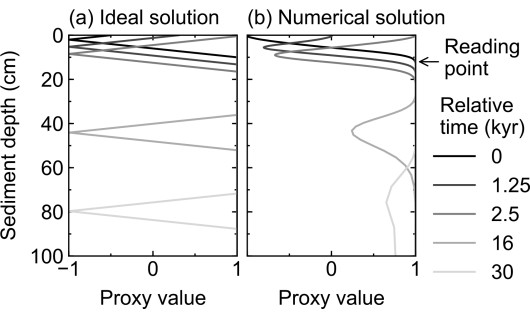

**Figure 4.** Comparison of ideal **(a)** and numerical **(b)** solutions for burial advection of the proxy signal. To minimize the effect of numerical diffusion in numerical solution, signal values are read at just below the mixed layer as denoted by an arrow.

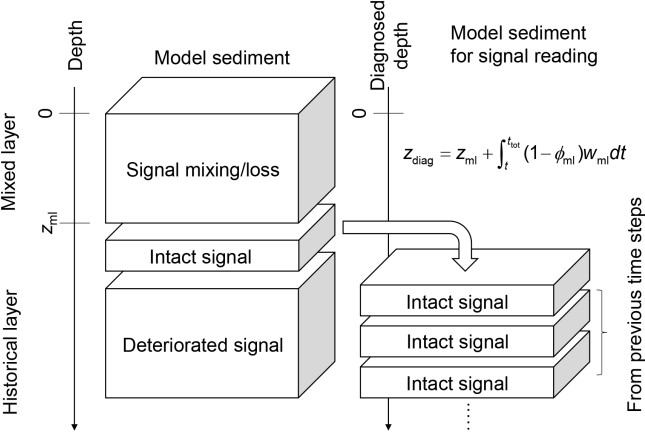

**Figure 5.** Schematic of the sediment column for signal tracking. The left side of diagram shows the sediment calculation domain that can be divided into mixed and historical layers. Signals are bio-mixed or lost by dissolution in the mixed layer and deteriorated at deep depths in the historical layer by numerical diffusion. The right side of diagram shows the sediment column for signal tracking which is composed of sediment layers that used to be located just below the mixed layer in the calculation domain and preserve proxy signals relatively well. Sediment depth in the latter system is denoted as "diagnosed depth" which can be calculated by the equation in the diagram or Eq. (27).

atively coarse (Fig. 4). An alternative is to allow for a partial surface layer and to accrete or remove complete layers depending on the growth or erosion at the surface, such as in Ridgwell (2007b). However, such an approach is impractical if the depth-dependent diagenetic reactions are to be solved rather than just recording historical accumulation (or erosion).

Here, to minimize the effect of numerical diffusion, we read out the proxy signal as a function of time, from just below the mixed layer and before the start of the "historical" layer ($z_{ml}$, see arrow in Figs. 4 and 5). Accordingly, signal values are not plotted against the depth of the sediment domain but against a sediment stack composed of the sediment

layers that were used to record the proxy signal (i.e., at depth $z_{ml}$) during the course of the simulation. The depth of this sediment stack is called diagnosed depth ($z_{diag}$, Fig. 5) and can be calculated as follows:

$$z_{diag} = z_{ml} + \int\limits_{t}^{t_{tot}} (1 - \phi_{ml}) w_{ml} \, dt, \tag{27}$$

where $\phi_{ml}$ and $w_{ml}$ ($\mathrm{cm\,yr^{-1}}$) denote the porosity and burial velocity at the mixed-layer depth ($z = z_{ml}$), respectively, and $t_{tot}$ is the total duration of a simulation (years). While reading proxy signals at the bottom of the mixed layer is likely effective in most cases (cf. Supplement), it is also possible to specify a different depth point to read proxy signals. In such a case, the definition of diagnosed depth needs to be modified by replacing $z_{ml}$, $\phi_{ml}$ and $w_{ml}$ in Eq. (27) with the corresponding parameter values at the specified depth.

To convert the signal profiles plotted against diagnosed depth to profiles plotted against model time, an age model is required, which can be obtained by tracking model time as a proxy. The application of the three methods explained in Sect. 2.4.1 (i.e., to assign numerical values to multiple classes of $CaCO_3$ particles and calculate their rain fluxes from the input values) is not limited to tracking proxy signals but can also be applied to any other characteristic including the model time at which particles are deposited. In method 1, individual classes of $CaCO_3$ particles are defined based on the time steps discretized from a signal change event (Fig. 3a) and, thus, already have their own model time to be assigned with. Note, however, that tracking model time with method 1 is computationally more expensive because a larger number of explicit $CaCO_3$ classes is needed to represent the continuously changing model time. When using method 2 or 3 to track model time in addition to paleoceanographic proxies, the number of $CaCO_3$ classes must be doubled (cf. Eq. 26). For example, when using method 2, one proxy signal can be simulated with two (or a pair of) $CaCO_3$ classes representing the maximum and minimum proxy value. Additionally tracking model time requires an extra pair of $CaCO_3$ classes, whereas the start and end of model time is assigned to the two pairs, respectively (cf. Eq. 26). In either method, model time tracked in bulk $CaCO_3$ can be plotted against diagnosed depth, which is the age model of IMP, and can be used to plot the other tracked proxy signals against model time. Examples of obtaining and using IMP's age model are provided in the Supplement.

## 3 Results and discussion

### 3.1 Diagenesis

In this section, we highlight diagenetic aspects of the model including comparison with the $CaCO_3$ diagenesis model by Archer (1991).

First, the capability of the model to obtain steady-state and time-dependent sediment profiles of solid and aqueous species is illustrated by showing a spin-up phase and a transient phase between two steady states, respectively, of a simulation. We then compare lysoclines estimated by IMP and the diagenesis model of Archer (1991). The lysocline is the ocean depth below which $CaCO_3$ dissolution significantly increases, and the depth of the lysocline is an important indicator for determining the Earth's carbon cycle response to environmental changes (e.g., sea level change) and associated feedbacks on climate (e.g., Archer and Maier-Reimer, 1994; Ridgwell et al., 2003; Ridgwell and Zeebe, 2005; Munhoven, 2007; Greene et al., 2019). $CaCO_3$ dissolution below the lysocline is caused because the thermodynamic stability of $CaCO_3$ decreases due to increased pressure, but the lysocline is also known to be significantly affected by local rain fluxes of OM and $CaCO_3$, and early diagenesis within sediments (e.g., Archer, 1991). Therefore, simulating the depth of the lysocline is a good test of a $CaCO_3$ diagenesis model. The details of the experiments and results are described in the following subsections.

### Experimental setup

To illustrate the initial evolution of the model, a spin-up experiment was run until a steady-state sediment composition was achieved. For this, we assumed Fickian mixing using the default conditions given in Table 1 (Fig. 1a). Model output includes depth profiles of density and volume fraction of solid sediment (Fig. 6a, c), burial velocity (Fig. 6b), concentrations of solid and aqueous species (Fig. 6d–k), and rates of biogeochemical reactions (Fig. 6l–n) for five time instances during the spin-up experiment (1, 10 and 100 kyr, and 1 and 3 Myr).

A second experiment illustrates how a change in the boundary conditions affects the temporal evolution of the depth profiles in IMP. This experiment starts from the end of the first spin-up experiment and artificially imposes significant carbonate dissolution by changing the water depth from 3.5 to 5.0 km between 5 and 45 kyr (Fig. 7). Because of the longer timescale to achieve steady state (see the first experiment), the second experiment run for 50 kyr is in transient states except for the initial steady state at 0 kyr (Fig. 7).

Finally, IMP was run to steady state assuming various carbonate rain fluxes (ranging from 6 to 60 $\mathrm{\mu mol\,cm^{-2}\,yr^{-1}}$, in increments of 6 $\mathrm{\mu mol\,cm^{-2}\,yr^{-1}}$), ratios of organic matter to carbonate (0, 0.5, 0.67, 1 and 1.5) and water depths (ranging from 0.24 to 6.00 km, in increments of 0.24 km) (cf. Archer, 1991). These lysocline experiments were performed for both the oxic-only OM degradation model and the oxic–anoxic model (Figs. 8, 9). To facilitate comparison of our results with Archer (1991), IMP assumes a single class of $CaCO_3$ particles, Fickian mixing for bioturbation and a sediment column depth of 50 cm. All other boundary conditions are as described in Table 1.

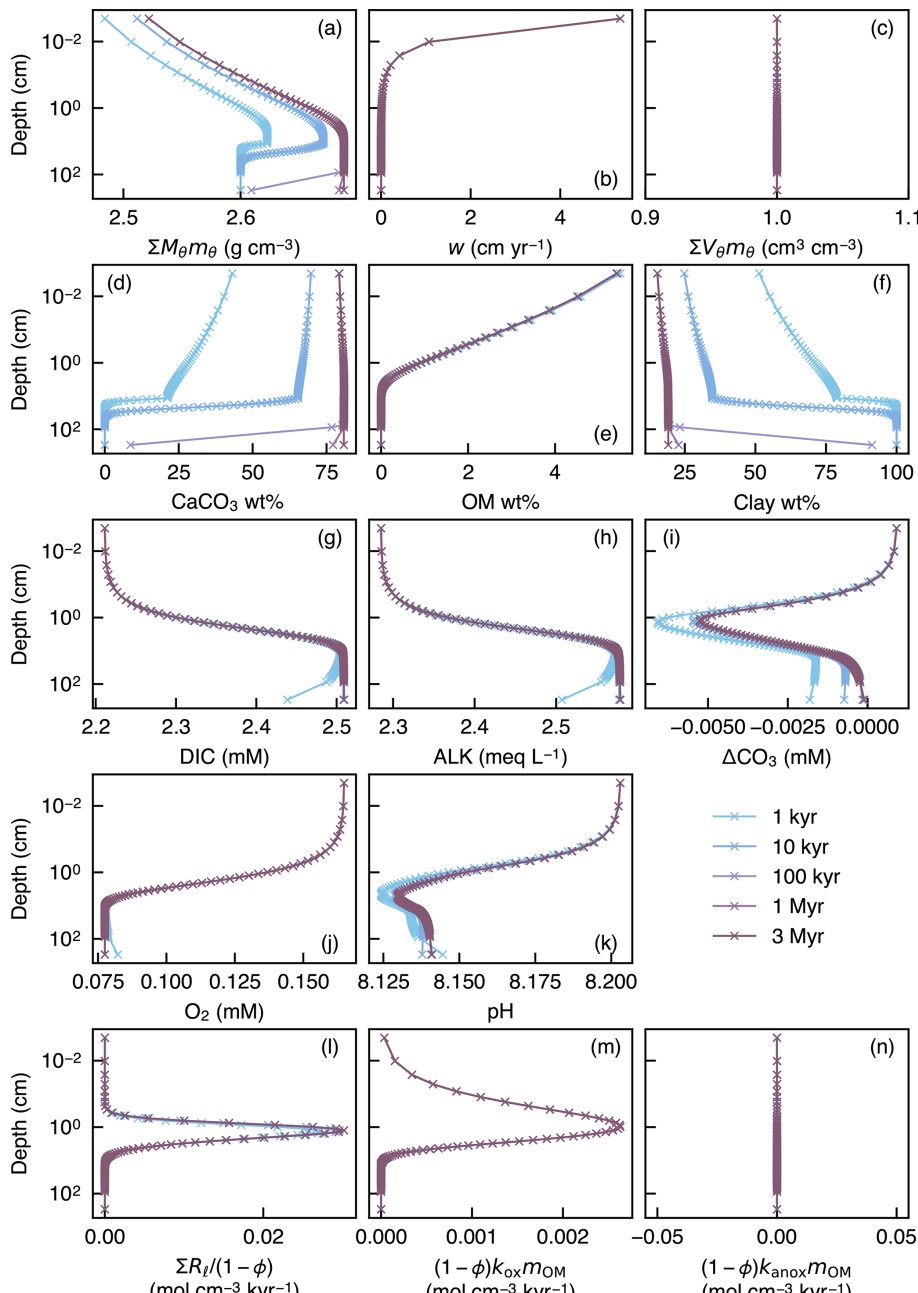

**Figure 6.** Depth profiles of the density **(a)** and volume fraction **(c)** of solid sediment, burial velocity **(b)**, weight fractions of bulk $CaCO_3$ **(d)**, organic matter **(e)** and nonreactive detrital materials **(f)** in solid sediment, porewater concentrations of total dissolved $CO_2$ species **(g)**, carbonate alkalinity **(h)** and oxygen **(j)**, deviation of porewater carbonate concentration from that in equilibrium with $CaCO_3$ **(i)**, porewater pH **(k)**, dissolution rate of $CaCO_3$ **(l)**, and decomposition rate of organic matter in the oxic **(m)** and anoxic **(n)** zone of sediment, as a function of time. The boundary conditions of the model are parameterized with the default parameter values (Table 1). The calculations assume four classes of $CaCO_3$ particles and Fickian mixing for bioturbation. Illustrated is the temporal evolution of the depth profiles from initial conditions (Sect. 2.3) to a steady state.

One can use the IMP code of any of the three programming languages (i.e., Fortran90, MATLAB or Python) to conduct the simulations presented in this paper. The model code for each language is stored in the respective directory (i.e., "Fortran", "MATLAB" and "Python"), and a language-specific readme file provides instructions on how to run the simulations (e.g., `\iMP\Fortran\readme_Fortran.txt` for the Fortran version). The boundary conditions can be specified with time-invariant values at run time (e.g., the third experiment above; see the readme file for the chosen ver-

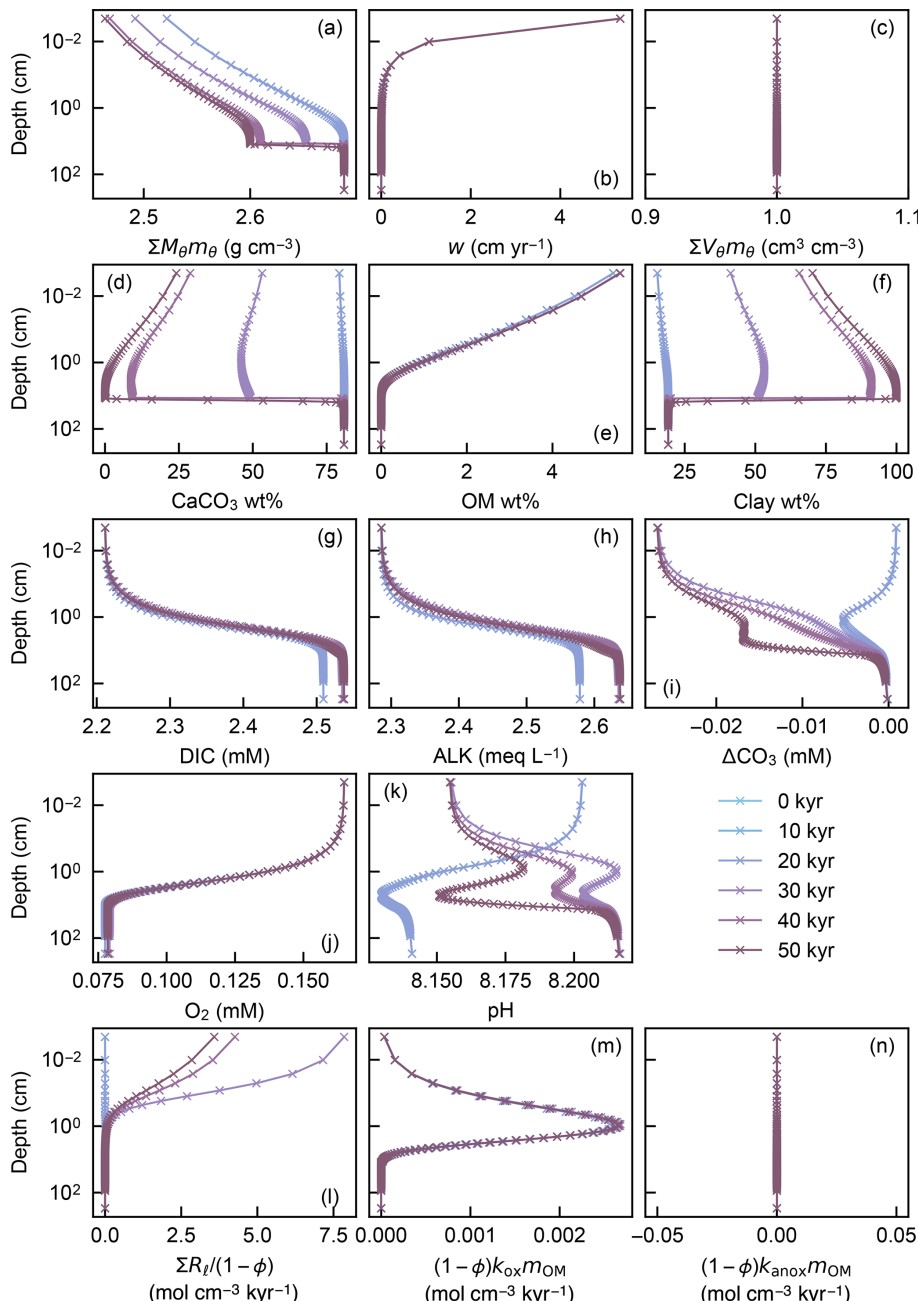

**Figure 7.** Depth profiles of the density (**a**) and volume fraction (**c**) of solid sediment, burial velocity (**b**), weight fractions of bulk $CaCO_3$ (**d**), organic matter (**e**) and nonreactive detrital materials (**f**) in solid sediment, porewater concentrations of total dissolved $CO_2$ species (**g**), carbonate alkalinity (**h**) and oxygen (**j**), deviation of porewater carbonate concentration from that in equilibrium with $CaCO_3$ (**i**), porewater pH (**k**), dissolution rate of $CaCO_3$ (**l**), and decomposition rate of organic matter in the oxic (**m**) and anoxic (**n**) zone of sediment, as a function of time. The boundary conditions of the model change with time as in dissolution experiment 2 (Sect. 3.2.2, Fig. 12). The calculations assume four classes of $CaCO_3$ particles and Fickian mixing for bioturbation. Illustrated are the temporal evolutions of the depth profiles which are initially at steady state at 3.5 km of water depth but perturbed by water depth change to 5.0 km between 5 and 45 kyr.

sion of the code) but can also be changed as a function of time (as in the second experiment above). The temporal changes in the boundary conditions must be prescribed in the input files that are stored in a directory "input" and can be modified by the user (see the readme file therein,

`\iMP\input\readme_input.txt`, for the details). We also provide Python scripts to plot concentrations of solid and aqueous species (e.g., Figs. 6–9) as well as tracked proxy signals (Sect. 3.2), stored in a directory "plot" (see a readme

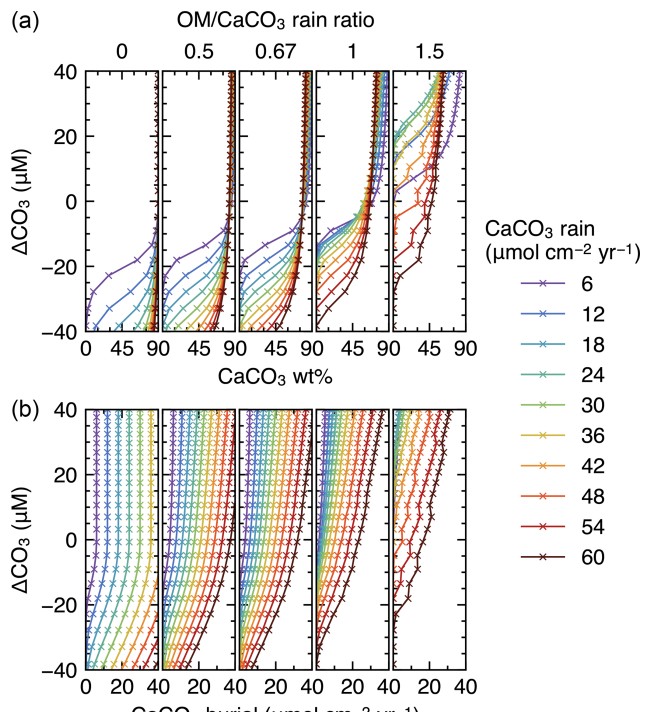

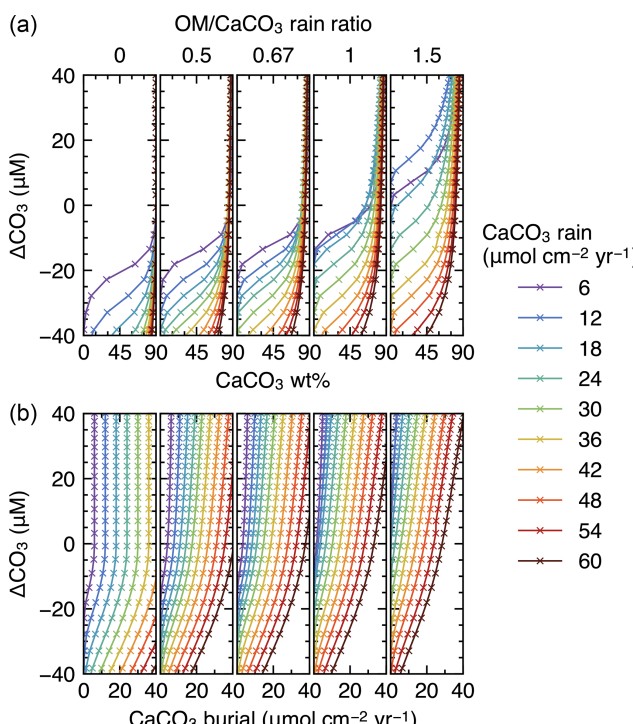

**Figure 8.** Estimated $CaCO_3$ weight fractions in mixed-layer **(a)** and burial fluxes **(b)** as functions of the $CaCO_3$ saturation degree and rain fluxes, with only oxic degradation of organic matter enabled. Saturation degree is measured by the difference of the carbonate ion concentration at the seawater–sediment interface from that at calcite saturation, $\Delta CO_3$. The results shown are from the model with a shallower sediment depth (50 cm) and single class of $CaCO_3$ particles.

**Figure 9.** Same as Fig. 8 but enabling both oxic and anoxic degradation of organic matter.

file therein, `\iMP\plot\readme_plot.txt`, for more details).

## Results

In the spin-up to steady state, spaces for solid sediment defined by assumed porosity $(1 - \phi)$ are initially empty (not filled) because of the low initial concentrations of solid species $(\sum_\theta V_\theta m_\theta \cong 0$; Sect. 2.3) but soon become filled with clay (as a "dilatant") and OM, and later with $CaCO_3$ as Eq. (22) is enforced and steady state is approached $(\sum_\theta V_\theta m_\theta = 1$; Fig. 6a, c). In contrast, pore spaces are assumed to be always filled with pore water and pore-water chemistry achieves steady state much faster (Fig. 6g–k) (e.g., Archer et al., 2002). The steady-state results for bulk phases (Fig. 6) are not affected by changing the number of $CaCO_3$ classes or the time step of each time integration (cf. Sect. 2.3.2).

The second experiment demonstrates that once steady state is achieved, a change in boundary conditions does not generate significant void spaces $(\sum_\theta V_\theta m_\theta \ll 1)$ and/or expansions $(\sum_\theta V_\theta m_\theta \gg 1)$ in solid sediment (Fig. 7c), thus

generally satisfying Eq. (22). In other words, prescribed spaces for solid sediment by assumed porosity are almost perfectly matched with the sums of volumes of all solid-phase species $(\sum_\theta V_\theta m_\theta = 1$; Fig. 7c) even when the concentrations of solid species dynamically change with time, leaving steady state (e.g., Fig. 7d). Absence of significant void spaces or expansions in solid sediment provides a convergence diagnostic (adapted from one of the convergence diagnostics in the steady-state diagenesis model of Archer et al., 2002).

Finally, we compare steady-state lysoclines simulated with IMP to results from the $CaCO_3$ diagenesis model of Archer (1991), who showed that the lysocline is sensitive to rain rates of carbonate and organic matter to the seafloor and, in particular, to the ratio of these fluxes. The simulated lysocline and carbonate burial rates for the oxic-only OM degradation model are presented in Fig. 8a and b. The results for the oxic–anoxic model are shown in Fig. 9a and b.

In general, our predicted mixed-layer $CaCO_3$ wt % and the $CaCO_3$ burial fluxes match the steady-state estimates by Archer (1991) (compare with Figs. 5 and 6 from Archer, 1991). For instance, as in Archer (1991), increasing the carbon rain to the sediments for lower $OM/CaCO_3$ rain ratios (i.e., $\leq 0.67$) enhances carbonate preservation and causes the lysocline to deepen for both the oxic-only and the oxic–anoxic OM degradation model (Figs. 8, 9). The only notable difference occurs for the oxic-only OM degradation model

under the most extreme carbon rain fluxes (i.e., rain ratio = 1.5; $CaCO_3$ rain > 40 $\mu$mol cm$^{-2}$ yr$^{-1}$). Here, IMP simulates higher $CaCO_3$ preservation than the model of Archer (1991) (Fig. 8, right panels). This difference can be explained by a burial velocity enhancement caused by high organic matter preservation in the oxic-only model, which is not considered by Archer (1991) (see the lysocline experiment with $V_{OM} = 0$ in the Supplement). For the same high OM/$CaCO_3$ rain ratio (1.5), the oxic–anoxic OM degradation model simulates an enhancement in the carbonate accumulation rate and a deepening of the lysocline for an increase in the $CaCO_3$ rain, which is in line with the results of Archer (1991).

## 3.2 Signal tracking diagenesis

In the following subsections, we illustrate the utility of the model for exploring the combined effects of bioturbation and chemical erosion on the preservation of proxy signals in carbonates. The experiments presented here adopt method 2 for the signal and flux assignment (Fig. 3), as it is a more accurate and computationally less expensive approach than method 1 and is more flexible than method 3 (Sect. 2.4.1). Equivalent results using methods 1 and 3 are described in the Supplement to demonstrate that all methods lead to the same results.

All experiments simulate two paleoceanographic proxies simultaneously, $\delta^{13}$C and $\delta^{18}$O, and both proxy signals change over the course of the experiments in an idealized fashion. All experiments adopt the oxic–anoxic OM degradation model and, if not stated otherwise, the default conditions in Table 1. Signal values are plotted against diagnosed depth (see Fig. 5 and Eq. 27). The same series of experiments as in Sect. 3.2 but tracking model time in addition to $\delta^{13}$C and $\delta^{18}$O are presented in the Supplement, where we illustrate that proxy signal values can be plotted against model time using the model specific age model (Sect. 2.4.2).

### 3.2.1 Bioturbation

#### Experimental setup

The effects of three different styles of bioturbation on the recorded proxy signals are considered: (i) Fickian local mixing with a biodiffusion coefficient of $D_{b,\theta} = 0.15$ cm$^2$ yr$^{-1}$, (ii) homogeneous nonlocal mixing to represent random mixing as simulated by, e.g., TURBO2 (Trauth, 2013) and (iii) process-based nonlocal mixing simulated by deposit-feeder automata from the LABS model (e.g., Boudreau et al., 2001; Choi et al., 2002; Kanzaki et al., 2019). Because the LABS-derived transition matrix contains less continuous and more irregular transport probability than the other two styles of bio-mixing (Fig. 1), it is susceptible to convergence problems (cf., Boudreau, 1997, Sect. 2.2.2). When convergence was not achieved, model results with bio-mixing from LABS are not shown in the following subsections (Sects. 3.2.1–3.2.3).

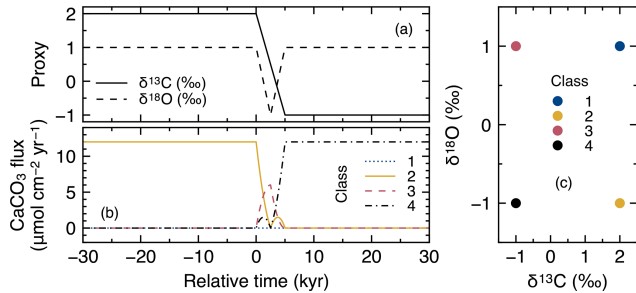

**Figure 10.** Timelines of proxy inputs (**a**) and rain fluxes of individual classes of $CaCO_3$ particles (**b**) with different proxy values (**c**) in simulations examining signal distortion by bioturbation.

The input proxy values of $\delta^{13}$C and $\delta^{18}$O in $CaCO_3$ either experience a step change over 5 kyr or a 5 kyr duration impulse event, respectively (Fig. 10a). Four end-member classes of $CaCO_3$ particles are used for signal tracking (Fig. 10c), and simulated proxy signals are recorded just below the sediment mixed layer and plotted against diagnosed depth to minimize the effect of numerical diffusion (Sect. 2.4.2). A first set of experiments is conducted with dissolution disabled for all $CaCO_3$ classes ($k_{cc,\ell} = 0$) in order to solely consider the effect of different styles of bioturbation. In a second set of experiments, the default $CaCO_3$ dissolution rate constant is used for all classes.

## Results

To visualize signal distortions by comparison, the input signals as a function of time (Fig. 10a) are plotted against diagnosed depth in Fig. 11, using the age model for the no bioturbation case (Supplement). Slight deviations of the recorded signals (pink curves in Fig. 11a and b) from the input signals (dotted black lines) in the "no bioturbation" case can be attributed to numerical diffusion but are minor compared with signal distortions exhibited by bioturbated sediments (blue, yellow and green curves). More specifically, dispersion of the recorded signals occurs over a larger depth interval and, for the impulse event in $\delta^{18}$O, the signal magnitude is significantly reduced with bioturbation (Fig. 11a, b). Fickian and homogeneous mixing distorts the input signals similarly (blue and yellow curves, respectively, which are almost completely superimposed in Fig. 11a and b), but LABS mixing results in slightly different signal shifts that extend to deeper depths (green curves). This difference may be explained by defecating/pushing of particles by deposit-feeder automata resulting in rare occasions where particle displacements propagate to depths even below the mixed layer (Fig. 1c; e.g., Choi et al., 2002). Note that bio-mixing in LABS can vary with assumed physicochemical and ecological conditions and animal types (e.g., Boudreau et al., 2001; Kanzaki et al., 2019); thus, our results should not be

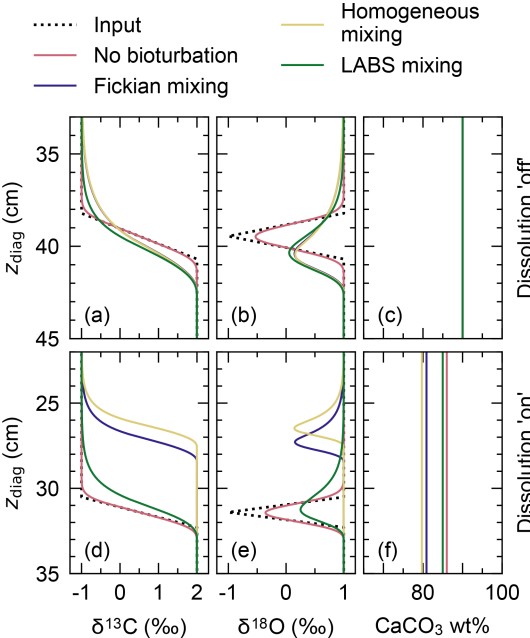

**Figure 11.** Proxy signals **(a, b, d, e)** and weight fraction of bulk CaCO$_3$ in solid sediment **(c, f)** tracked by four classes of CaCO$_3$ particles plotted against diagnosed depth in simulations examining signal distortion by bioturbation. In panels **(a)**–**(c)**, dissolution rate constants of all CaCO$_3$ classes are fixed at zero, whereas in panels **(d)**–**(f)**, they are fixed at the default value (Table 1).

regarded as the exclusive results with a LABS transition matrix (cf. Sect. 2.2.2).

Results for the second set of experiments with CaCO$_3$ dissolution enabled are presented in Fig. 11d–f. Different modes of bioturbation result in variations in the extent of CaCO$_3$ dissolution (Fig. 11f): no bioturbation leads to the lowest degree of dissolution, and efficient homogeneous mixing causes the highest degree of dissolution (Fig. 11f). Correspondingly, sediment accumulation rates and, thus, age models differ between different styles of bioturbation (Supplement), and one observes signal change events at shallower depths with a more enhanced dissolution (Fig. 11d, e). By enabling dissolution, proxy signals are slightly lost along with CaCO$_3$ particles, especially when bio-mixing is not efficient. This can be recognized by a reduction in the magnitude of the $\delta^{18}$O impulse for the no bioturbation case by enabling dissolution (slightly smaller peak of the pink curve in Fig. 11e than in Fig. 11b). We examine the dissolution effect in more detail in the next subsection.

### 3.2.2 Dissolution of carbonates

**Experimental setup**

While evidence for significant dissolution of sedimentary carbonates provides information about ocean chemistry (e.g., Oxburgh and Broecker, 1993; Zachos et al., 2005; Panchuk

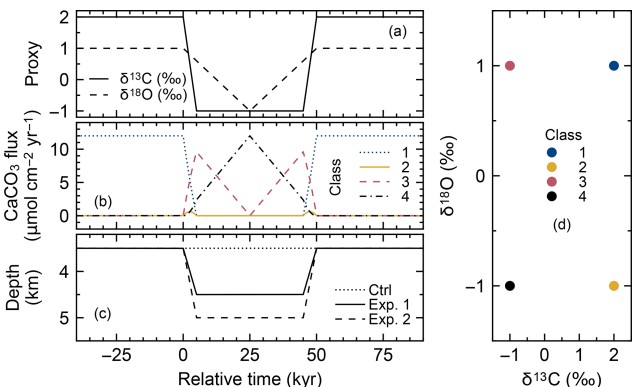

**Figure 12.** Timelines of proxy inputs **(a)**, rain fluxes of individual classes of CaCO$_3$ particles **(b)** with different proxy values **(d)** and water depth changes **(c)** in simulations examining signal distortion by CaCO$_3$ dissolution. Two different water depth changes are considered, denoted as dissolution experiments 1 and 2 **(c)**. One set of experiments was conducted without changing the water depth for comparison (dotted line in **c**).

et al., 2008), it also distorts proxy signals recorded in these carbonates. In this subsection, we examine how and to what extent dissolution distorts proxy signals.

We consider a negative $\delta^{13}$C excursion over 40 kyr with a relatively rapid onset and recovery of the isotope signal (over 5 kyr). At the same time, a more gradual ramp down and up change of the $\delta^{18}$O signal over 50 kyr is simulated (Fig. 12a). The signal shifts for the two proxies are intentionally made decoupled in time and should not be associated with any "real" geological event. These signal changes are accompanied by water depth changes from the background depth of 3.5 to 4.5 and 5.0 km over 5 kyr in order to cause different extents of dissolution (Fig. 12c) through destabilizing CaCO$_3$ by increasing pressure (Millero, 1995). These imposed changes in water depths are not intended to be "realistic"; rather, they drive conditions of enhanced CaCO$_3$ dissolution as might have been caused by environmental changes such as ocean acidification (e.g., see: Ridgwell, 2007b), but without the additional interpretative complications of actually changing the ocean chemistry at the sediment surface in the model. (Note that it is also possible to drive IMP with changing upper geochemical boundary conditions to explicitly simulate, e.g., ocean acidification.) The water depth and related dissolution changes are assumed to be synchronous with the proxy signal changes (Fig. 12a, c).

Signal tracking is conducted by simulating the same four classes of CaCO$_3$ as in the previous subsection (Fig. 12d; cf. Fig. 10c), with enabling Fickian or homogeneous biomixing (Fig. 1a, b) or without bioturbation. An additional set of experiments was run without changing the water depth as a "no dissolution" control (dotted line in Fig. 12c). Simulated signals against sediment depth (Fig. 13) are compared with input signals (dotted black curves in Fig. 13) which are

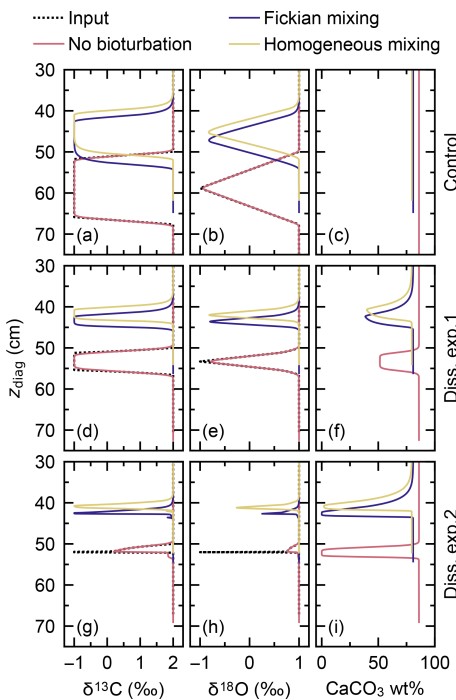

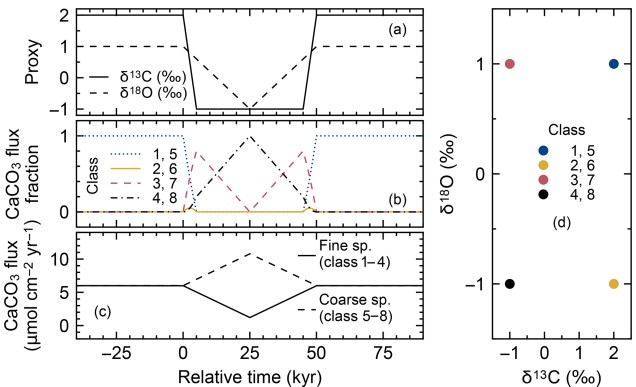

**Figure 14.** Timelines of proxy inputs **(a)**, normalized rain fluxes of individual classes of CaCO$_3$ particles **(b)** with different proxy values **(d)** and total rain fluxes of fine- and coarse-sized CaCO$_3$ species **(c)** in simulations examining the effect of species-specific mixing/dissolution properties. In panel **(b)**, rain fluxes of individual classes of fine and coarse CaCO$_3$ species are normalized against the total rain fluxes of respective fine and coarse CaCO$_3$ species from panel **(c)**.

**Figure 13.** Proxy signals **(a, b, d, e, g, h)** and the weight fraction of bulk CaCO$_3$ in solid sediment **(c, f, i)** tracked by four classes of CaCO$_3$ particles plotted against diagnosed depth in simulations examining signal distortion by CaCO$_3$ dissolution. Two different water depth changes are considered, denoted as dissolution experiments 1 and 2, and compared to the case without water depth change, denoted as control. See Fig. 12c for the assumed water depth changes.

obtained from their temporal changes (Fig. 12a) and the age model for the no bioturbation case (cf. Supplement) as in the previous subsection.

## Results

When dissolution is intensified by changing the water depth from 3.5 to 4.5 km (experiment 1; solid line in Fig. 12c), the total amount of CaCO$_3$ is reduced from ∼ 90 wt % to ∼ 50 wt % for all cases with and without bioturbation (Fig. 13f). As described in Sect. 3.2.1, dissolution is enhanced by bio-mixing, and signal change events are correspondingly observed at different depths between different modes of bioturbation (Fig. 13d–f; cf. Supplement). Apparent durations of the signal change events become shorter compared with the control experiment (Fig. 13a–c) because less sediment accumulates during the events with more enhanced dissolution (Fig. 13c, f). However, because imposed dissolution is still moderate (Fig. 13f) and relatively long-term signal change events are considered (e.g., compare Fig. 12a with Fig. 10a), no significant reduction in the magnitude of signal peaks is observed in experiment 1.

Further increasing the dissolution rate by changing the water depth to 5.0 km during the isotope excursion (experiment 2; dashed line in Fig. 12c) causes CaCO$_3$ to completely disappear for all cases with and without bioturbation (Fig. 13i). Note that a concentration of absolute zero is not allowed for solid species in the model. Simulated concentrations are truncated at a threshold of $10^{-300}$ mol cm$^{-3}$. As for dissolution experiment 1 (Fig. 13f), different styles of bioturbation cause different CaCO$_3$ dissolution rates (Fig. 13i). Under this more intense dissolution scenario, simulated proxy signals are considerably distorted and reduced for all styles of bioturbation (Fig. 13g, h). Simulated excursions of proxy signals are observed for considerably shorter apparent duration or sediment depth interval, as described in the paragraph above.

It is noted that the carbon and alkalinity fluxes from dissolved CaCO$_3$ in sediments under any destabilization can vary with the mode of bioturbation (Figs. 11, 13). This indicates the potential role of benthic ecosystems to determine the feedback of sedimentary CaCO$_3$ to a climate perturbation (e.g., Ridgwell, 2007b; Jennions et al., 2015).

### 3.2.3 Species-specific mixing/dissolution

#### Experimental setup

It has been suggested that carbonates of different sizes can be differently bioturbated and dissolved in marine sediments (e.g., Broecker et al., 1991; Bard, 2001; Barker et al., 2007). IMP is well suited for examining the effect of differential mixing and/or dissolution rate among CaCO$_3$ size classes on the signal distortion.

**Table 4.** Properties of CaCO$_3$ classes for simulations in Sect. 3.2.3.

| Property* | CaCO$_3$ class | | | | | | | |
|---|---|---|---|---|---|---|---|---|
| | 1 | 2 | 3 | 4 | 5 | 6 | 7 | 8 |
| $\delta^{13}$C (‰) | 2 | 2 | −1 | −1 | 2 | 2 | −1 | −1 |
| $\delta^{18}$O (‰) | 1 | −1 | 1 | −1 | 1 | −1 | 1 | −1 |
| Size[a] | Fine | Fine | Fine | Fine | Coarse | Coarse | Coarse | Coarse |

\* Coarse classes have the default values for the dissolution rate constant and bio-mixing parameters in Table 1. Fine classes have a 10 times higher dissolution rate constant and a 20 cm mixed-layer depth, but the parameter values are otherwise the same as the coarse classes. TS2

Here, we consider eight CaCO$_3$ classes, consisting of two sets of the same four CaCO$_3$ classes as in the previous subsections (Table 4). We assign two distinctive sizes to these two sets (Fig. 14c, d). CaCO$_3$ particles in the first set are assumed to be of "fine" grain size and are consequently bioturbated (by Fickian and in a second experiment by homogeneous mixing) to deeper depths (20 cm; cf., Bard, 2001) with the correspondingly modified transition matrices (Eqs. 18 and 19; Sect. 2.2.2). They are also dissolved at a faster rate by adopting a dissolution rate constant increased by a factor of 10 (cf., Keir, 1980) (classes 1–4 in Fig. 14 and Table 4). CaCO$_3$ particles in the second set are of "coarse" grain size and adopt the default particle characteristics (Table 1, classes 5–8 in Fig. 14 and Table 4) and transition matrices (Fig. 1a, b). The total mass flux and isotope signal input are the same as in Sect. 3.2.2, and the water depth remains unaltered at 3.5 km. In concert with the $\delta^{18}$O decrease, the coarse species becomes more dominant over the fine species (the rain fraction of the coarse species increases from 50 % to 90 %; Fig. 14c; cf., Schmidt et al., 2004).

**Results**

The differences in dissolution and mixing properties of fine and coarse CaCO$_3$ species have a prominent effect on their relative preservation (Fig. 15c). In general, the coarse species shows higher preservation due to its lower dissolution rate. The more efficient the adopted mixing mode (e.g., homogeneous mixing), the better the preservation of the coarse species and the more obscured the preservation of the imposed CaCO$_3$ input flux changes. Correspondingly accumulation rates are different for fine and coarse CaCO$_3$ species; thus, excursions of proxy signals as well as peaks in coarse vs. fine species abundance are offset by ∼ 10 cm between the two species (compare solid and dotted curves in Fig. 15). Observed apparent offsets of peaks in proxy signals and species abundance can be mostly removed by applying individual age models to the two species, although the reduction in the magnitude of abundance shifts cannot be recovered (Supplement).

Although the above experiment is not designed to simulate any specific surface-environment change event in the past, signal offsets among CaCO$_3$ species have been observed in,

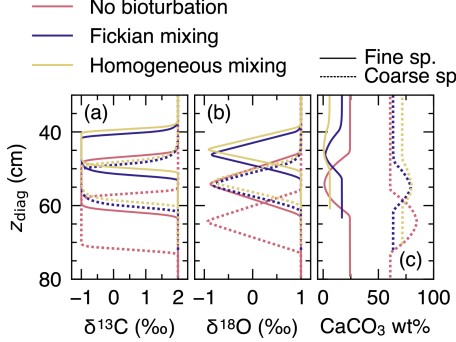

**Figure 15.** Proxy signals **(a, b)** and the weight fraction of bulk CaCO$_3$ in solid sediment **(c)** for fine and coarse CaCO$_3$ species (solid and dotted curves, respectively) tracked by eight classes of CaCO$_3$ particles in simulations examining the effect of species-specific mixing/dissolution properties.

e.g., hyperthermal events (e.g., Kirtland Turner et al., 2017). The application of IMP to such events can be useful, as it might lead to an insight into population shifts among calcifiers associated with environmental changes in the past (cf. Figs. 14 and 15c).

## 3.3 Proxy signals in an extended environmental parameter space

The complexity of IMP also allows for hypothesis testing that has not been possible with traditional diagenetic models. For instance, changes in the rain fraction of fine vs. coarse species in the signal tracking experiment in Sect. 3.2.3 affected proxy signals of both species differently. However, in traditional 1-D diagenetic models, such an environmental variable is not explicitly considered. This section reiterates the utility of IMP to interpret proxy signals in a parameter space that is not accessible when considering only bulk CaCO$_3$. Here, we focus on $^{14}$C age as another example proxy.

In equatorial Pacific sediments, carbonate $^{14}$C ages have been observed to increase with decreasing CaCO$_3$ wt % (circles in Fig. 16), a counterintuitive trend if CaCO$_3$ is dissolved homogeneously, as dissolution should shift the distribution towards younger CaCO$_3$ particles. Although Broecker

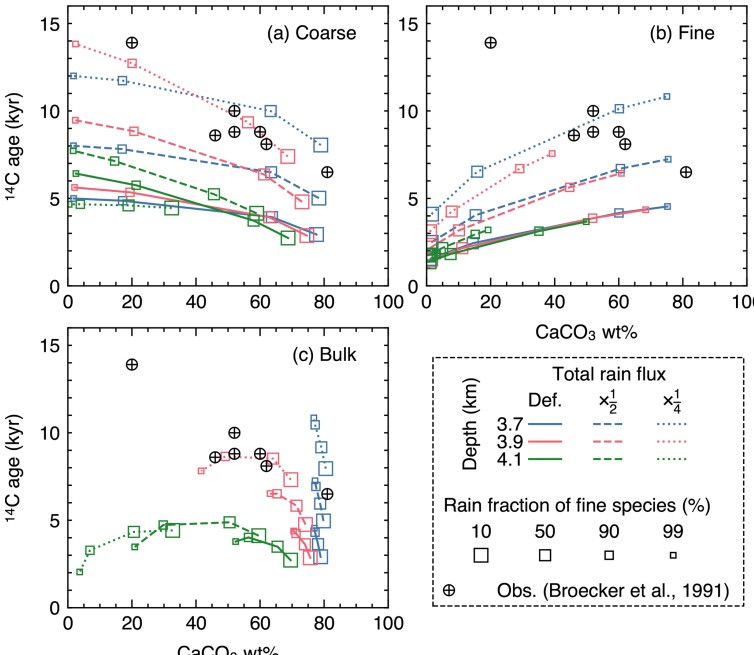

**Figure 16.** Radiocarbon ages plotted against $CaCO_3$ wt % in the mixed layer for **(a)** coarse and **(b)** fine $CaCO_3$ species, and **(c)** bulk $CaCO_3$. The values at 12 cm sediment depth are assumed to represent those in the mixed layer.

et al. (1991) demonstrated with an idealized sediment box model that interface dissolution ($CaCO_3$ dissolution completed before bio-mixing and burial) can reproduce the observation, the mechanism does not allow $CaCO_3$ dissolution to continue within the sediment column and, thus, cannot be implemented by 1-D reactive-transport models, which usually assume homogeneous dissolution (cf. Keir, 1984; Keir and Michel, 1993; Broecker et al., 1991). However, it is not known whether homogeneous dissolution can lead to a different $^{14}$C age vs. $CaCO_3$ wt % relationship in a more complicated and, thus, realistic parameter space, especially where distinct $CaCO_3$ size classes are explicitly accounted for. Here, we simulate steady-state $^{14}$C age in the mixed layer for the coarse and fine species considered in Sect. 3.2.3.

**Experimental setup**

To track radiocarbon age, the direct tracking method (method 3 in Sect. 2.4.1) is utilized. The method simulates five $CaCO_3$ classes corresponding to five isotopologues ($Ca^{12}C^{16}O_3$, $Ca^{12}C^{18}O^{16}O_2$, $Ca^{13}C^{16}O_3$, $Ca^{13}C^{18}O^{16}O_2$ and $Ca^{14}CO_3$) to track four associated isotopic signals ($\delta^{13}$C, $\delta^{18}$O, $\Delta_{47}$ and $^{14}$C age) recorded in $CaCO_3$ particles that are of the same size (see the Supplement for the details). Because we further track the "size" of $CaCO_3$ particles by simulating two distinct $CaCO_3$ species of "fine" and "coarse" sizes, two sets of the above five classes (i.e., 10 classes in total) are necessary (cf. Eq. 26; Table 5). The first set of five classes (classes 1–5) possesses the dissolution and bio-mixing properties for the

fine species defined in Sect. 3.2.3, whereas the second set (classes 6–10) represents the coarse species (Table 5).

Steady-state simulations were run with the above 10 $CaCO_3$ classes adopting Fickian mixing, for three water depths (3.7, 3.9 and 4.1 km), three total sediment fluxes (12 (default), 6 and 3 μmol total $CaCO_3$ cm$^{-2}$ yr$^{-1}$ with the fixed default OM/$CaCO_3$ and clay/$CaCO_3$ rain ratios; Tables 1, 2) and for different rain fractions of the fine $CaCO_3$ species (10 %, 50 %, 90 % and 99 %). The rain fluxes of individual $CaCO_3$ classes are calculated from the total $CaCO_3$ rain, the rain fraction for fine species, and assuming that $\delta^{13}$C = $\delta^{18}$O = $\Delta_{47}$ = 0‰ and the $^{14}$C/$^{12}$C ratio is $1.2 \times 10^{-12}$ (Aloisi et al., 2004) (cf. Supplement). The mixed-layer depth (and, thus, the transition matrix) and the dissolution rate constant are defined differently between the fine (classes 1–5) and coarse (classes 6–10) species (cf. Sect. 3.2.3). All the other parameters were set at the default values (Table 1).

**Results**

Because dissolution and transport is fully coupled in IMP (i.e., dissolution is "homogeneous"), a decrease in $CaCO_3$ wt % caused by increasing water depth generally leads to a younger radiocarbon age (e.g., see Fig. 16c where $^{14}$C ages are highest for blue curves and lowest for green curves). However, when the decrease in the $CaCO_3$ concentration is caused not by increasing the water depth but by increasing the rain fraction of the fine species that dissolves faster (trajectories depicted with curves in Fig. 16), the trend of $^{14}$C age vs. $CaCO_3$ wt % differs. The trend for the coarse

**Table 5.** Properties of CaCO$_3$ classes for simulations in Sect. 3.3.

| Property | CaCO$_3$ class | | | | | | | | | |
|---|---|---|---|---|---|---|---|---|---|---|
| | 1 | 2 | 3 | 4 | 5 | 6 | 7 | 8 | 9 | 10 |
| Comp. (Ca–)[a] | $^{12}C^{16}O_3$ | $^{12}C^{18}O^{16}O_2$ | $^{13}C^{16}O_3$ | $^{13}C^{18}O^{16}O_2$ | $^{14}CO_3$ | $^{12}C^{16}O_3$ | $^{12}C^{18}O^{16}O_2$ | $^{13}C^{16}O_3$ | $^{13}C^{18}O^{16}O_2$ | $^{14}CO_3$ |
| Size[b] | Fine | Fine | Fine | Fine | Fine | Coarse | Coarse | Coarse | Coarse | Coarse |

[a] Isotopologue composition of each CaCO$_3$ class, denoted without Ca. [b] Coarse classes have the default values for dissolution rate constant and bio-mixing parameters in Table 1. Fine classes have a 10 times higher dissolution rate constant and a 20 cm mixed-layer depth, but the parameter values are otherwise the same as the coarse classes.

species is especially counterintuitive, where an older $^{14}C$ age is observed for lower CaCO$_3$ wt % (Fig. 16a). The opposite trend is recognized for the fine species (Fig. 16b). Bulk CaCO$_3$ shows a combination of the above two contrasting aging trends, and whether bulk $^{14}C$ age increases or decreases with bulk CaCO$_3$ wt % depends on the contribution of fine vs. coarse species (Fig. 16c). The magnitude of the aging effect (whether by changes in the rain fraction of the fine species or the water depth) can be amplified when the total sediment rain is decreased because both CaCO$_3$ species are buried at a slower rate (dashed and dotted curves in Fig. 16).

Note that it is not our intention to perfectly reproduce the observations with the parameterization adopted in this experiment, given that a large number of parameters would need to be constrained and/or modified (e.g., Keir, 1980; Walter and Morse, 1984, 1985; Bard, 2001). Nonetheless, the $^{14}C$ age sensitivity to the rain fraction of fine species shown above illustrates the utility of the model to interpret proxy signals in an extended and more realistic environmental parameter space.

## 4 Conclusions and summary

Our new Implicit model of Multiple Particles (diagenesis) – IMP – is capable of tracking proxy signals by implicitly simulating reactive transport of multiple solid carbonate particles, along with calculations of organic matter, refractory detrital materials, and aqueous oxygen and dissolved CO$_2$ species. The model also realizes simulations of different kinds of bioturbation by adopting different transition matrices. As shown with illustrative experiments, signal distortion can vary with the style of bioturbation, intensity of chemical erosion and distributions of CaCO$_3$ species with different dissolution/mixing characteristics. Such complexity needs to be carefully evaluated when reading proxies in marine sedimentary carbonates for reconstruction of past environmental changes.

Future developments of the model include coupling with Earth system models, which will provide synthetic sedimentary records that are process based and can be directly compared with geological records. Coupling the model with an efficient Earth system model such as "cGENIE" (Ridgwell and Hargreaves, 2007; Ridgwell, 2007b) is particularly promising, as it may allow iterative runs to predict environ-

ment changes that minimize the difference between synthetic and observed sedimentary records (e.g., Kirtland Turner and Ridgwell, 2013).

*Code availability.* The IMP source codes are available on GitHub (https://github.com/imuds/iMP) under the MIT License. The specific version used in this paper is tagged as "v1.0" and has been assigned a DOI (https://doi.org/10.5281/zenodo.5213875, Kanzaki and Hülse, 2021). A readme file on the web provides the instructions for executing the simulations.

*Data availability.* The observational data shown in Fig. 16 are available in the paper cited in the figure legend CE2.

*Supplement.* The supplement related to this article is available online at: https://doi.org/10.5194/gmd-14-1-2021-supplement.

*Author contributions.* YK designed and implemented the model in Fortran90 with contributions from the other authors. DH and YK converted the Fortran90 version to MATLAB and Python versions, respectively. YK designed the simulations with contributions from the other authors. All authors contributed to writing the paper.

*Competing interests.* The authors declare that they have no conflict of interest.

*Acknowledgements.* We are grateful to David Archer, Guy Munhoven and an anonymous reviewer for their useful comments on the paper and to Andrew Yool for the editorial handling. This research was supported by the Heising–Simons Foundation through a grant to Andy Ridgwell, Sandra Kirtland Turner and Lee Kump. Dominik Hülse was partially supported by the Simons Foundation.

*Financial support.* This research has been supported by the Heising-Simons Foundation (grant no. 2015-145) and the Simons Foundation (grant no. 653829).

*Review statement.* This paper was edited by Andrew Yool and reviewed by David Archer, Guy Munhoven and one anonymous referee.

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

**Remarks from the language copy-editor**

CE1    Please note the slight change to the requested edit.

CE2    Please confirm the edits.

**Remarks from the typesetter**

TS1    According to our standards, changes like this must first be approved by the editor, as data have already been reviewed, discussed and approved. Please provide a detailed explanation for those changes that can be forwarded to the editor. Upon approval, we will make the appropriate changes. Thank you for your understanding.

TS2    According to our standard, letters for footnotes are only used if there is more than one. For one footnote, an asterisk is used.