# Peer review of "A model for marine sedimentary carbonate diagenesis and paleoclimate proxy signal tracking: IMP v1.0"

_Geoscientific Model Development, 2020_

## Referee Comment (RC3)

Review of

**A model for marine sedimentary carbonate diagenesis and paleoclimate proxy signal tracking: IMP v0.9**

submitted to *Geoscientific Model Development*
by Y. Kanzaki et al.

**1  General comments**

Yoshiki Kanzaki and co-authors present the "Implicit model of Multiple Particles (and diagenesis)" (IMP). IMP is an early diagenesis model, build along the lines of the classical model of Archer (1991), which is extended to explicitly include the vertical distributions of all the solids considered. Three different solids are currently considered in IMP: carbonate, organic matter and clay (or any other inert material). Carbonate can be considered in multiple classes. IMP was developed to systematically explore the distortion of proxy signals (e.g., $\delta^{13}C$, $\delta^{18}O$,...) by bioturbation and chemical erosion. Different bioturbation models are considered: biodiffusion, homogeneous mixing, automaton based parametrisation derived from (e)LABS.

The paper fits well into the scope of *Geoscientific Model Development*. I found the study very interesting. The paper is generally well readable, although one stumbles here and there upon sentences whose meaning is not clear or whose syntax is not correct her. It would also gain a lot from a more precise language usage. These – minor – shortcomings should nevertheless be straightforward to fix.

There are other shortcomings that are far more important. The paper has been submitted as a *Model Description Paper*. The model description part is, however, of very uneven quality: while some parts are pleasantly detailed others are lacking even the most important information. It is, e.g., not explained how the $O_2$ penetration depth $z_{ox}$ is calculated.

The limitations of the diagenesis model are not discussed, not even mentioned. Sulfate reduction is the only sub-oxic/anoxic OM oxidation pathway, thus skipping the energetically more favourable $NO_3^-$ reduction, Mn(IV) and Fe(III) reduction pathways. No secondary redox reactions are considered. Some discussion on the implications of these simplifications would be of order.

Similarly, the strengths and disadvantages of the different bioturbation model formulations are not discussed (the biodiffusive approach leads to block tridiagonal Jacobians, that can be inverted by an efficient block oriented Thomas algorithm, whereas the LABS derived transition matrix is likely to be full, without any special structure) and thus contributing to a Jacobian that will be computationally costly to invert.

Finally, the experiment descriptions also leave too many questions open. Here, especially the species-dependent mixing experiment is poorly documented. The current text makes it impossible to understand how exactly the model has been set-up for this experiment.

I am convinced that this manuscript can make an interesting contribution to *Geoscientific Model Development*, provided it undergoes a major revision, to provide a better description of what is done, how it is done and why it is done that way.

**2 Specific comments**

**2.1 Model Basics and Structure**

**Page 8, Eq. (23):** Equation (23) is only valid for intraphase biodiffusion; for interphase bioturbation, there is an extra term related to the porosity gradient (see, e.g. Munhoven, 2021)

**Grid:** The grid description is unfortunately not clear. It starts with a language problem: "[…] discretized into $N = 100$ irregular grids where the grid size increases […]" does not make sense. Reading this as "[…] discretized into $N = 100$ irregular grid intervals where the interval size increases […]", and using the information provided in Table 2 (mapping function and control parameter value) to generate the underlying distribution (with 101 grid points delimiting 100 grid intervals), and paying attention to avoid catastrophic numerical cancellation in the factor $(\beta - 1)$, with $\beta = 1 + 5 \cdot 10^{-11}$, one obtains a very odd result: the deepest grid interval takes more than 81% of the total extent (405.8 cm), the second deepest 2.86% (14.3 cm) and the third one 1.68% (8.41 cm). So, provided the reported information on the grid generation is accurate, I would consider the bottom of the grid as essentially useless. The resulting extreme numerical diffusion might possibly deteriorate the quality of all the results. The extreme jump in the grid interval lengths by a factor of almost 30 from the second deepest

to the deepest interval (compared to less than 2 from the third to the second deepest layer) will lead to tremendous numerical diffusion, which might significantly influence the results further up in the sediment column. Although the adopted mapping function from the normalized regular to the finally used irregular grid is continuous (and continuously differentiable) – a condition that irregular grid mappings have to fulfil to ensure that consistency and convergence order of the numerical discretization schemes remain unchanged – the difference in the grid interval lengths acts, from a practical point of view more, like a discontinuity, which might ruin the convergence properties of the algorithm.

One might furthermore wonder why the model grid has to extend to a depth of 5 m. The useful ("undisturbed" information is tapped at a much shallower depth, just below the mixed layer (i.e., typically at 10–20 cm depth). With a calcite/clay input mass ratio of about 9:1, at most of the order of 10 cm of sub-mixed-layer sediment can possibly be unburied during a chemical erosion event and the information stored in the gridded part of the sediment column deeper than, say 20–30 cm is actually not required here..

I finally also wonder whether the recorded "undisturbed" signal would be consistent with the signal recorded in the main sediment column, even if there were no numerical diffusion – numerical diffusion was put forward as a reason for tapping the useful information right below the mixed layer: even in the absence of numerical diffusion, the signal in the main column may be altered by sub-mixed-layer chemical reactions. Furthermore, if chemically altered material later becomes subject to chemical erosion (i.e., returns to the mixed-layer), its composition can be expected to be different from that in the "undisturbed" record. How important might the resulting inconsistencies be?

**Page 10, lines 267–273:** Why not move this transition matrix representation of the biodiffusive model of bioturbation to Sect. 2.2.2 ("Bioturbation")? In that place it would contribute to emphasize the overarching nature of the transition matrix approach. A priori, it was not clear in my mind that biodiffusion could also be part of the transition matrix approach (albeit in discretized form).

**Section 2.5.1 (pages 10–11):** This is obviously the central part of this manuscript. I find it quite short in that respect. The advantages/disadvantages (potentials/shortcomings) of the different methods could be stated in more

detail. I would also like to challenge the authors on several assertions made here. I do not see why method 3 is less flexible than methods 1 and 2. To me, methods 2 and 3 are mathematically speaking absolutely identical. The only difference I can see is in the way the input data have to be processed. That processing can, however, be considered to be carried out outside the model itself.

**Page 12, lines 332–345:** The discussion about how time tracking could be implemented is, as far as I can see, not entirely correct. Obviously, method 1 would be extremely costly to adopt for time, but it clearly is the one that would offer the best time resolution. Method 1 essentially discretizes the complete age-dimension carried by one sedimentary component (e.g., calcite) as an ever-growing number of $CaCO_3$ variants, each one representing one age-class. I is difficult to comprehend how method 2 could possibly conserve more accurate information with only two variables, compared to method 1. For any given time-step, the input data for method 1 are given by one single class, and those for method 2 by one interpolated value between the end members. To me, one appears as accurate as the other, for any given single time-step. For the next time step, another single class is created and added for method 1, while another interpolated value between the end members is added for method 2, and so forth. After $n$ time steps with method 1, one can clearly distinguish between the fates of each single class of material brought in during the $n$ steps; with method 2, this is not possible, as the $n$ interpolates have been added, with little chance to deconvolve the resulting information. So, to me, method 1 is the accurate one, not method 2.

**Page 14, lines 410–412:**

> "This difference can be explained by a burial velocity enhancement caused by high organic matter preservation in the oxic-only model, which is not considered by Archer (1991)."

This is probably correct, but why not check it instead of speculating? This can be easily done by setting $V_{OM}$ to 0.

**2.2 Experiments and Results**

**Throughout:** there seems to be a mismatch between "time step" and "instants in time" Actually, no information about the step lengths (time steps) adopted for the integration of Eq. (1) in time.

**Page 14, lines 398–400:**

> "Absence of significant void spaces or expansions in solid sediment has been adopted as a convergence diagnostic by the sediment diagenesis model of Archer et al. (2002), [...]"

This sentence could possibly be misunderstood, as this is actually not the only diagnostic used by Archer et al. (2002). Their convergence diagnostic is first "[...] based on conservation of mass to within 2% for all solid phase and dissolved constituents (appropriate balances between rain, reaction, diffusion, and burial rates)." (Archer et al., 2002, p. 17-4, par. [19]). They continue writing that the sum of the solid phase concentrations *also* provides a convergence diagnostic.

Conserving mass to within a few % only – a side-effect of requiring letting the sum of the solids' volume fractions float within a few % (line 255, page 9)) – might be sufficient for steady-state calculations. In transient simulation experiments, I would anticipate that deviations of the order of a few % could cause considerable model drift in transient simulation experiments. It would be better not to leave any loose ends and therefore to enforce strict static volume conservation.

The statement at line 400 ("The results of the second experiment thus confirm [...]") is a non sequitur. The results do not confirm the applicability of the model for time-dependent simulation experiments: they only do not infirm it.

**Page 15, lines 437*ff***:  Although I understand that multiplying the transition matrix by 1/10 might facilitate the calculation, it also reduces the mixing intensity by a factor 10, and thus describes a completely different setting. Are the results for the so reduced mixing intensity accepted as final in case the calculations converge, or are they used as a starting point for a continuation method, wherein the mixing intensity is then gradually increased to approach the original matrix? I think that the results obtained with the matrix divided by 10 cannot be used for a comparison with others where no such reduction was adopted.

**Section 3.2.3, pages 17–18**  This experiment needs to be better documented. Critical information is missing here: what bioturbation model has been used to get these results? In the text, we read about different bioturbation depths for small and large particles, with reference to Bard (2001). The

bioturbation model of Bard (2001) is, mathematically speaking, rather convoluted and it is not clear to me how it might be transposed to the transition matrix framework adopted here. At first sight, it is possibly a homogeneous mixing model, but I am not entirely even sure that it is possible to transpose it at all into the transition matrix framework of IMP. In that model, the bioturbation depth is the main control parameter. How that bioturbation depth has been used as a control parameter here – if it has, which is unclear as well – is not explained. Please make this description more complete and if necessary also provide additional theoretical foundations

It should also be noticed that some theoretical homework is of order when it comes to different mixing rates for small and large particles. The solids' advection rate is actually always defined as the movement rate of the bulk solids (Meysman et al., 2005, see, e.g.). Accordingly, the advective rate may not be equal to $w$ in Eq. (1) – this is already not the case when interphase biodiffusion is adopted as a bioturbation model. In case a biodiffusion model is used, it should be noticed that Eq. (23) is not applicable (this is mentioned as a sufficient condition at lines 214–216 for Eq. (23) to hold, but it is also necessary).

**3 Technical and minor corrections**

**Abstract:** Please include the meaning of the model name acronym somewhere in the abstract.

**Page 3, lines 65–66:** "The reason for this is that published sediment mixing models are generally unable to realize diagenetic reaction" – not sure what is meant here (should "are generally unable to realize" possibly read "do generally not take into account"?)

**Page 3, line 81:** "Following presentation" should read "Following the presentation" (or "After the presentation" ?)

**Page 4, line 91:** Is it possible to "share" *distinct* characteristics?

**Page 6, line 160:** "mocsy" should read "mocsy 2.0"

**Pages 6 and 7:** The notation $dz_i$ for the thickness of layer $i$ is not recommendable, as there are later ratios of such thicknesses are used (e.g., page 7, Eqs. (13) and (14)), which could be confused with derivatives and thus lead to unnecessary misunderstandings. I suggest to replace $dz_i$ by $\delta_i$ or $\Delta_i$, perhaps $h_i$.

**Page 7, line 181:** Is $A$ the horizontal cross-sectional area?

**Page 7, lines 195–196:** Very cumbersome and syntactically incorrect sentence. I suggest to reformulate it as "Equation (15) is a finite difference version of Eq. (17) ..."

**Page 8, lines 212 and 213:** I suggest to replace $D_b$ by $D_{\theta,b}$ or something the like in htese two instances to emphasize that different biodiffusion coefficient values may be used for different particle classes (different classes of particles are later supposed to be transported in different ways by bioturbation) The special case for a single $D_b$ for all solids then comes more naturally at line 214.

**Page 9, lines 236–237:** This sentence may be misleading, as "specified at the beginning of each time integration" could be wrongly interpreted as saying that an time-explicit approach is used in IMP

**Page 9, lines 253:** How is "significantly different" translated quantitatively?

**Page 10, lines 277*ff*:** Wich version of LABS was used here in the end? Reed et al. (2007)? Or was it eLABS (Kanzaki et al., 2019)? Please specify.

**Page 10, Equation system (24):** First of all, this way of defining $(K_{\theta,ij})$ is difficult to understand. At first it looks like some kind of implicit definition. Is there not more clear way to write this?
   Second there seem to be two errors:

- "$2 \leq j = i + 1 = n_{ml}$" should probably read
  "$2 \leq j = i + 1 \leq n_{ml}$"

- "$1 \leq j = i - 1 = n_{ml} - 1$" should probably read
  "$1 \leq j = i - 1 \leq n_{ml} - 1$"

**Page 11, line 309:** "[...] and accompanied generation of alkalinity, [...]": not sure what this could mean. With $Ca^{14}CO_3/CaCO_3$ ratios of the order of $10^{-14}$, alkalinity changes by $Ca^{14}CO_3$ decay should really be on the negligible side of life.

**Page 11, lines 294 and 298:** "$2^{n_P}$" should read "$2n_P$" as each proxy requires two end-members.

**Page 13, lines 364–365:** "5 time steps" should most probably read "five instants in time". By the way, which time step was chosen for the integration? A variable one? a constant one – how long?

**Page 13, lines 377_ff_:** It would be fair to state that these are replications of experiments from Archer (1991).

**Page 14, line 392–393:** Strange sentence. – please reformulate.

**Page 14, line 410:** "than Archer (1991) model" would more correctly read "than the model of Archer (1991)"

**Page 14, line 434:** "provability" should probably read "probability"

**Page 14, line 413:** "in $CaCO_3$ rain" should read "of the $CaCO_3$ rain"

**Page 15, line 439:** "are now shown" should read "are not shown", I guess.

**Page 16, line 455:** would "at depths" not better read "from depths"?

**Page 16, lines 470 and 472:** Text imprecise: chemical erosion requires dissolution, but dissolution does not necessarily lead to chemical erosion. Please reformulate.

**Page 17, line 490:** "When dissolution is imposed [...]"? Would "When dissolution is intensified [...]" not be more correct?

**Page 17, lines 503–504:** "Simulated proxy signals are considerably shorter in apparent duration as described in the above paragraph." – not sure what this means.

**Page 18, line 520:** "the more" should read "the better"

**Page 18, line 521:** "accumulation rate differs between" would better read "accumulation rates are different for"

**Page 18, line 539:** "The source codes of IMP model" should read "The IMP source codes" (delete "model")

**Page 18, line 540:** "specific version used of the model" should read "specific model version used"

**Page 22, lines 636–637:** This URL points to the secondary JSTOR archive copy of the reference. It better had to be replaced by the DOI of the original paper (available in open access): DOI:10.5670/oceanog.2009.100

**Page 24, line 710:** The DOI of the MATLAB version 1.1 of CO2SYS is not resolving any more. The current URL is https://cdiac.ess-dive.lbl.gov/ftp/co2sys/CO2SYS_calc_MATLAB_v1.1.

**Page 39, Table 1:**

- I guess, "Number of sediment grids" means "Number of sediment grid points" as there is most probably only one grid.

- For the density of OM, a value of $1.2\,\mathrm{g/cm^3}$ is reported with reference to Mayer et al. (2004). I have not been able to find that value of $1.2\,\mathrm{g/cm^3}$ in Mayer et al. (2004). Considering the $\rho_{OM}$ values reported for marine samples in Table 1 of that reference, I find a higher value of $1.45 \pm 0.23\,\mathrm{g/cm^3}$. Please clarify.

It would be good to specify more clearly that $OM \equiv CH_2O$. Only in this case some of the ratios such as the OM:$CaCO_3$ ratio $r$ make sense, as one mole of OM then represents one mole of OC. Readers used to Redfield composition might be confused else.

**Figures 2, 7, 8, 10:** It is recommended not to use green and red/orange colours tones in parallel on a graph (see https://www.geoscientific-model-development.net/submission.html - "Figures & Tables", point 7)

**References**

D. Archer, J. L. Morford, and S. R. Emerson. A model of suboxic sedimentary diagenesis suitable for automatic tuning and gridded domains. *Global Biogeochem. Cy.*, 16(1):1017, 2002. doi: 10.1029/2000GB001288.

D. E. Archer. Modeling the calcite lysocline. *J. Geophys. Res.*, 96(C9):17037–17050, 1991. doi: 10.1029/91JC01812.

E. Bard. Paleoceanographic implications of the difference in deep-sea sediment mixing between large and fine particles. *Paleoceanography*, 16(3): 235–239, 2001. doi: 10.1029/2000PA000537.

Y. Kanzaki, B. P. Boudreau, S. Kirtland Turner, and A. Ridgwell. A lattice-automaton bioturbation simulator with coupled physics, chemistry, and biology in marine sediments (eLABS v0.2). *Geosci. Model Dev.*, 12(10): 4469–4496, 2019. doi: 10.5194/gmd-12-4469-2019.

L. M. Mayer, L. L. Schick, K. R. Hardy, R. Wagai, and J. McCarthy. Organic matter in small mesopores in sediments and soils. *Geochim. Cosmochim. Ac.*, 68(19):3863–3872, 2004. doi: 10.1016/j.gca.2004.03.019.

F. J. R. Meysman, B. P. Boudreau, and J. J. Middelburg. Modelling the reactive transport in sediments subject to bioturbation and compaction. *Geochim. Cosmochim. Ac.*, 69(14):3601–3617, 2005. doi: 10.1016/j.gca.2005. 01.004.

G. Munhoven. Model of Early Diagenesis in the Upper Sediment with Adaptable complexity – MEDUSA (v. 2): a time-dependent biogeochemical sediment module for Earth system models, process analysisand teaching. *Geosci. Model Dev.*, 14(6):3603–3631, 2021. doi: 10.5194/ gmd-14-3603-2021.

D. C. Reed, B. P. Boudreau, and K. Huang. Transient tracer dynamics in a lattice-automaton model of bioturbation. *J. Mar. Res.*, 65(6):813–833, 2007. doi: 10.1357/002224007784219039.

Liège, 15th June 2021

Guy Munhoven

---

## Author Comment (AC1)

**Response to Referee #1 (Dr. David Archer)**

We express our gratitude to Dr. David Archer for his useful comments. Our response to the reviewer's comments and the corresponding revision are described in detail and separately below. The numbers of pages, lines, equations, tables and figures are those in the revised manuscript unless otherwise described.

Comment 1:

"The paper could be much more interesting with the addition of some additional sensitivity experiments. The authors cite the literature on the 14-C distribution in CaCO3 in the equatorial Pacific, which gets older as the %CaCO3 goes down, the opposite of what you would expect if the CaCO3 dissolved homogeneously – a low %CaCO3 would imply a short residence time and a low age. Somehow shells become "armored" from dissolution if they survive 1rly stage. I have attempted to replicate this by using multiple phases of CaCO3 with varying dissolution kinetics or solubilities, but I never managed to reproduce the trend in the observations. I think this observation is mirrored in the 14-C age distribution of mollusk shells; it seems to be a general thing. Perhaps attacking this problem is for a future study, while this is just a model development paper, but it doesn't seem like another sensitivity plot or two would add too much baggage to the paper."

Response:

We agree with the reviewer that including the potential model application to the 14-C age problem would make the paper more interesting. As suggested by the reviewer, we conducted a sensitivity analysis for the rain fraction of fine and coarse species considered in the experiment in Section 3.2.3. In these experiments we adopt method 3 instead of method 2 for tracking 14-C age. We vary water depth (from 3.7 to 4.1 km) and total sediment rain (from the default value of 12 μmol $CaCO_3$ $cm^{-2}$ $yr^{-1}$, to 6 and 3 μmol $CaCO_3$ $cm^{-2}$ $yr^{-1}$, with fixed $OM/CaCO_3$ and clay/$CaCO_3$ ratios, Fig. 16).

We found that the trend of increasing $^{14}C$ age with lower $CaCO_3$ wt% can be simulated for a coarse $CaCO_3$ species by increasing the rain fraction of the fine species (Fig. 16a). On the other hand, an opposite trend is found for fine species (Fig. 16b). The above aging effects for coarse and fine species are enabled because the fine species dissolves faster and increasing the rain fraction of the fine species leads to a longer residence time if total rain and water depth are fixed. Bulk $^{14}C$ age and $CaCO_3$ wt% shows a trend that is a combination of the opposing trends for the fine and coarse species (Fig. 16c). Therefore, whether bulk $^{14}C$ age decreases or increases with $CaCO_3$ wt% depends on the contribution of fine vs. coarse species along the trajectory for the fine rain fraction at a fixed sediment rain and water depth.

While we could not reproduce the observation perfectly, introducing the fine and coarse species can show the $^{14}C$ age vs. $CaCO_3$ wt% trend which is not possible when considering only bulk phases, thus supporting the utility of IMP for a better interpretation of proxy signals.

Changes in manuscript (Page numbers/Line numbers):
We added a sensitivity analysis described above as Section 3.3 (P20/L601-P22/L646) and Fig. 16 (P44).

Comment 2:
"I had some questions as I was reading, points of confusion. For the Fickian diffusion, does the rate taper off exponentially with depth or is it an abrupt cutoff? What is the difference between Ficking diffusion and homogeneous mixing?"

Response:
Fickian mixing is a 'local' mixing, where particle translocations occur only between adjacent layers. Homogeneous mixing is introduced as one example of 'non-local' mixing, where particles can be exchanged between remote layers.

   We did not implement the biodiffusion coefficient as a function of depth, although tapering off the coefficient with depth might be more realistic (e.g., Ridgwell, 2001, Glacial-interglacial perturbations in the global carbon cycle, PhD thesis). The simplified parameterization for the biodiffusion coefficient still serves our purpose, that is, to illustrate the effects of variation in bio-mixing styles on signal distortion.

Changes in manuscript (Page numbers/Line numbers):
We added an illustration of modified transition matrices $K_{\theta,ij}$ as Fig. 1 (P29) and more explanations on the difference between local and non-local mixing referring to Fig. 1 (P8/L229-P9/L241).

   We clarified that the biodiffusion coefficient for Fickian mixing does not change with depth (P8/L210-211).

Comment 3:
"line 45: clarify what you mean; I would have thought that Fickian diffusion is random mixing. In that section it might also be worth mentioning that some models use uniform mixing down to an abrupt cutoff, while others use an exponential dropoff in mixing rate."

Response:
Please see our response to comment 2 by the reviewer where we address the issue mostly.

   The point of this paper is to illustrate the model's capability to simulate the effects of changes in bio-mixing style on proxy signals, rather than those of changing the parameterization of Fickian mixing. So we mentioned the parameterization of Fickian mixing only in the later section (Section

2.2.2).

Changes in manuscript (Page numbers/Line numbers):
Please see our changes in manuscript in response to comment 2 by the reviewer.

Comment 4:
"line 110. Kudos for coding the model up in multiple languages! But why python, when Julia seems just as elegant and flexible and also lots faster?"

Response:
Julia is a younger language than Python and thus we assumed that Python user population could be larger than Julia user population. Python is slow but as in other languages it can call a module created from a Fortran code. We include such Python (plus Fortran) usage option and a readme file on our code repository (iMP/Python/readme_Python_Fortran.txt) instructs how to create a Python module from the Fortran code and call it from a Python script. Julia is definitely one of candidates which will be used in the future release of IMP.

Changes in manuscript:
No change was made in response to the comment.

Comment 4:
"line 225. How can the initial condition have vanishinly small concentrations of all of the solid phases? Don't they have to sum to fill the solid volume implied by the time independent porosity? (On further reading I understood that this is an initial state for an iteration, which by the time it converges will have solved the problem. However, maybe a sentence here would help clarify.)"

Response:
As stated in the parenthesis by the reviewer, small solid concentrations deficient for solid space prescribed by porosity are allowed only as an initial state of an experiment. Later time-integration fills up the initial void space and once filled there is no void solid space or expansion of solids compared to the prescribed solid space by porosity.

Changes in manuscript (Page numbers/Line numbers):
We revised manuscript to avoid potential confusion (P10/L275-276).

Comment 5:

"line 260. "time implicit method". It took a bit of digging to figure out if the model is time dependent or steady-state? This was a clue in the text but it didn't specify whether it applied to solid and dissolved species or what. I figured it out from the figures, but it would have been useful to state it more explicitly earlier on. (And on that, why bother with time dependence for the solutes? It must slow things down a lot.)"

Response:

The model is time dependent, as we stated earlier e.g., in lines 87-88. We agree that it would be better to clarify that time-dependent calculation was made for all species in the relevant sentence.

Although including the time dependent simulations slows the calculation, the applicability of the model increases. For example, one will probably need the time dependent simulations even for solutes when considering a diagenesis including deep reactions involving methane cycling (e.g., Archer, 2007, Biogeosciences 4, 521) although not considered in the simulations presented in this paper.

Changes in manuscript (Page numbers/Line numbers):

We revised the relevant sentence to be clearer that the model conducts time-dependent simulations (P11/L319).

Comment 6:

"Equations 23-24. These are succinct descriptions of the matrices, but they are not very transparent as far as explaining what the mixing models do. Why does homogeneous mixing use P rather than D? (On subsequent rereading there is an extensive discussion on the formulation of homogeneous mixing, but a bit of summary here would be helpful.)"

Response:

We agree that description of transition matrices was not very transparent in the previous manuscript.

Changes in manuscript (Page numbers/Line numbers):

We added the detailed description of transition matrices to Section 2.2.2 (P8/L229-P9/L241) and heat maps of transition matrices as Fig. 1 (P29).

Comment 7:

"Would it be possible to make some kind of visualization of the transportation matrix, a heat map of some sort that would show how the mixing mechanisms differ?"

Response:

We agree to provide heat maps of transition matrices to facilitate comparison between different styles of bio-mixing.

Changes in manuscript (Page numbers/Line numbers):

The heat maps of transition matrices were added as Fig. 1 (P29).

Comment 8:

"line 500. It would be interesting to integrate how much excess CaCO3 dissolution occurred due to the change in solubility (water depth) – how the buffering strength of the sea floor depends on the mixing model."

Response:

Bio-mixing affects $CaCO_3$ dissolution to a given destabilization (a water-depth increase), as can be inferred from simulated wt% $CaCO_3$ record (Fig. 13). This indicates that buffering strength would change with bio-mixing style. However, because we enforce dissolution in an idealized way, i.e., not in a realistic way in the experiments in Section 3.2.2, we avoid providing exact values of $CaCO_3$ dissolution fluxes.

Changes in manuscript (Page numbers/Line numbers):

We added description of the potential changes of buffering strength with bio-mixing styles (P19/L570-572).

Comment 9:

"line 515. Do the smaller particles have higher surface to volume, and also less mass, so they dissolve more quickly for those reasons also? It would be useful to add differences in kinetics or solubility here, and separate out the different effects."

Response:

Thermodynamic differences between fine and coarse species can be implementable in IMP as stated in lines 34-35 but not included in any experiments conducted in this paper. Kinetic differences between fine and coarse species are considered in the experiment in Section 3.2.3 as stated in lines 578-584.

Changes in manuscript (Page numbers/Line numbers):

No change was made in response to the comment.

---

## Author Comment (AC2)

**Response to Referee #2**

We express our gratitude to Referee #2 for his/her useful comments. Our response to the reviewer's comments and the corresponding revision are described in detail and separately below. The numbers of pages, lines, equations, tables and figures are those in the revised manuscript unless otherwise described.

General comment:
"I applaud the authors for providing such a well-revised, organized and thought-out manuscript. My critique is minimal and provided as a suggestion for two general areas of improvement: (1) expansion of explanation of transition matrices, namely automation-based and (2) expansion of the discussion."

Response:
We are grateful to the reviewer for appreciation of our work.
    Please see our response to specific comments by the reviewer below regarding suggestions (1) and (2).

Changes in manuscript (Page numbers/Line numbers):
Please see our changes in manuscript in response to specific comments below.

Specific comment 1:
"The manuscript would benefit from an expansion on the LABS simulation approach; automation-based transition matrices described in the methods could be more thorough. As written, the paper requires unfamiliar readers to investigate this approach outside of the paper. This could also be achieved in the introduction."

Response:
Agreed. Except that we prefer to add the details of the LABS approach to Section 2 rather than to the Introduction.

Changes in manuscript (Page numbers/Line numbers):
We added more explanation of the LABS simulations as well as the transition matrices to Section 2.2.2 (P8/L218-P9/L241).

Specific comment 2:

"While this model will be applied to interpreting archives of geologic events and such events are cited as motivation, there is little to no discussion later in the paper of the significance of their experiments with regard to these events. For example, what is the significance of the model result in which coarse fraction species become more dominant, in terms of records of past abrupt events of environmental change? Perhaps the authors decided to stay away from interpreting their model development results in terms of geologic applications, but some model-data comparison may be warranted in the discussion. This may or may not include a more representative simulation of an early Eocene hyperthermal event."

Response:

As a model development paper, we decided to stay away from detailed attempts to use the model for interpreting a specific geologic record. Nonetheless, general implications may be useful for the reader to have ideas about potential use of IMP. As an example, a geological event can be differently recorded by different biological species as recognized by e.g., offsets in the timing of isotopic excursions (e.g., Kirtland Turner et al., 2017). Such offsets between species might be able to be explained by IMP as illustrated in our example simulation where fine and coarse species are explicitly simulated with different dissolution and bio-mixing properties (Section 3.2.3).

Please note that we added Section 3.3 where a model-data comparison was made for 14-C age although we did not intend to fully reproduce the observations with the model (please see our response to specific comment 3 by the reviewer).

Changes in manuscript (Page numbers/Line numbers):
We mentioned the potential application of IMP in the description of the relevant simulation results (P20/L597-600).

Please also see our changes in manuscript in response to specific comment 3 by the reviewer.

Specific comment 3:
"Following this more generally, the discussion section of the manuscript is slightly limited and could be expanded. For example, how does this new model and the results of experiments in this study inform understanding of examples of processes outlined in the introduction? How might the findings here bias proxies in specific geologic archives (e.g., size fraction differences previously unaccounted for in proxy records)? This type of expansion would not necessarily require re-interpretation of records of e.g. the PETM, but rather clearly lay out the implications of their experiments which may be significant to a proxy-based researcher in the field."

Response:
We agree with the reviewer that more implications regarding interpretation of geological records can

be useful. The feature of IMP that species-specific records can be simulated will be useful for interpretation of the geological record where isotopic records are differently recorded by different biological species (e.g., Kirtland Turner et al., 2017). We further added a sensitivity analysis where 14-C age in the mixed layer is calculated as a function of the rain fraction of fine species, total sediment rain and water depth. This experiment was in parts motivated as the 14-C age problem (Broecker et al., 1991) is mentioned in the Introduction. The simulated relationships between the $^{14}$C-age and $CaCO_3$ wt% can differ between coarse and fine species and bulk $CaCO_3$ (Fig. 16). Our intention here is not to perfectly reproduce the observations, but the agreement between simulated and observed $^{14}$C trends (Fig. 16) illustrates the utility of IMP to explain proxy signals in a way not possible when considering only bulk $CaCO_3$.

Changes in manuscript:
We added a sensitivity analysis for 14-C age as Section 3.3 (P20/L601-P22/L646) and Fig. 16 (P44).

---

## Author Comment (AC3)

**Response to Referee #3 (Dr. Guy Munhoven)**

We express our gratitude to Dr. Guy Munhoven for his useful comments. Our response to the reviewer's comments and the corresponding revision are described in detail and separately below. The numbers of pages, lines, equations, tables and figures are those in the revised manuscript unless otherwise described.

General comment 1:
"The paper is generally well readable, although one stumbles here and there upon sentences whose meaning is not clear or whose syntax is not correct her. It would also gain a lot from a more precise language usage. These – minor – shortcomings should nevertheless be straightforward to fix."

Response:
We agree to revise the manuscript to make it more precise and readable reflecting the review comments. Please see our response to general/specific comments and technical/minor corrections by the reviewer below.

Changes in manuscript (Page numbers/Line numbers):
Please see our changes in manuscript in response to general/specific comments and technical/minor corrections by the reviewer below.

General comment 2:
"The model description part is, however, of very uneven quality: while some parts are pleasantly detailed others are lacking even the most important information. It is, e.g., not explained how the $O_2$ penetration depth $z_{ox}$ is calculated"

Response:
We did not elaborate upon the calculation of $z_{ox}$ because it has already been conducted by e.g., Emerson (1985) and Archer (1991). However, we agree with the reviewer that more details might be useful to the reader.

We calculate $z_{ox}$ together with OM and $O_2$ profiles iteratively. Specifically, the following steps are taken:
(1) $z_{ox}$ is calculated based on the $O_2$ profile from the previous iteration or time instance.
(2) OM and aerobic respiration profiles are calculated using $z_{ox}$ in step 1.
(3) the $O_2$ profile is calculated based on OM and aerobic OM respiration profiles through the following sub-steps:
(i) First, no diffusive flux is assumed as the lower boundary condition. If the resultant $O_2$ profile

satisfies $c_{O_2} > 0$ at all depth, then the $O_2$ calculation is finished.

(ii) If the above calculation results in $c_{O_2} \leq 0$ at any depth, a series of $O_2$ profile calculations are conducted with assuming $z_{ox} = z(i)$ where $i = 1$ to $N$ with the boundary condition of $c_{O_2} = 0$ at $z = z_{ox}$. Out of $N$ results, one where $c_{O_2}$ is closest to 0 at $z = z_{ox}$ is adopted.

(4) a new $z_{ox}$ is calculated based on the $O_2$ profile obtained in step 3.

(5) Steps 1–4 are repeated until $z_{ox}$ in steps 1 and 4 are located in the same sediment layer or both below the calculation domain.

Changes in manuscript (Page numbers/Line numbers):

We added explanations such as above on how OM and $O_2$ profiles are calculated iteratively (P11/L291-299).

General comment 3:

"The limitations of the diagenesis model are not discussed, not even mentioned. Sulfate reduction is the only sub-oxic/anoxic OM oxidation pathway, thus skipping the energetically more favourable $NO_3$ reduction, Mn(IV) and Fe(III) reduction pathways. No secondary redox reactions are considered. Some discussion on the implications of these simplifications would be of order."

Response:

As the reviewer pointed out, we did not address the effects of ignoring the OM decomposition by oxidants other than $O_2$ and $SO_4$, including $NO_3$, and Mn- and Fe-(oxyhydr)oxides, as well as relevant secondary reactions. We agree that the effects of omitting these reactions are important to mention/discuss.

Contribution of oxidants other than $O_2$ and $SO_4$ to OM decomposition is likely <~20% on the global scale according Archer et al. (2002) and Thullner et al. (2009). Therefore, OM decomposition by $O_2$ and $SO_4$ is a reasonable simplification of DIC and ALK fluxes from the OM reaction network. Although the current model cannot explicitly simulate OM degradation by oxidants other than $O_2$ and $SO_4$, an implicit implementation of an OM-decomposition-associated reaction is possible with IMP by adding DIC and ALK fluxes at a given depth although this option is enabled only for the Fortran version and is not used in this paper.

An example of usage of the above option is presented in Supplementary material, where the influence of DIC and ALK fluxes from anoxic oxidation of methane (AOM) in the deeper sediments on $CaCO_3$ diagenesis is simulated. Please note that this simulation adopted a different sediment grid structure ($z_{tot} = 200$ m) from that adopted in the main text ($z_{tot} = 5$ m) as AOM is assumed to occur at 10 m (please also see our response to specific comment 3 by the reviewer).

Changes in manuscript (Page numbers/Line numbers):

We added explanations to Section 2.2.1 so that the current model's limitation on OM degradation becomes clearer (P5/L143-145). Also, we mentioned that an implementation of ALK and DIC fluxes caused by a reaction that is not explicitly simulated in the model is possible (P5/L145-147). An application example of the above option was added to the Supplementary material (Section S1.2 in Supplementary material).

General comment 4:

"Similarly, the strengths and disadvantages of the different bioturbation model formulations are not discussed (the biodiffusive approach leads to block tridiagonal Jacobians, that can be inverted by an efficient block oriented Thomas algorithm, whereas the LABS derived transition matrix is likely to be full, without any special structure) and thus contributing to a Jacobian that will be computationally costly to invert."

Response:

We briefly mention the convergence difficulty when adopting LABS mixing and its cause in lines 432-437 of the previous manuscript. However, more details of the different mixing styles (or transition matrices) were not provided, which could be useful in order to discuss the numerical difference as inferred from the reviewer's comment. As pointed out by the reviewer, Jacobians are block tridiagonal with biodiffusion transition matrices, but not with homogeneous and LABS mixing as they are non-local mixing.

Transition matrices are represented with the transport probability $\tau P_{\theta,ij}$. We adopted a modified transition matrix $K_{\theta,ij}$, defined by Eq. (14), with which the difference equation of bio-mixing term becomes simple (Eq. (15)). Indeed, the matrix represented by components $(1 - \phi_i)K_{\theta,ij}$ at $(i, j)$ corresponds to the Jacobian matrix for the bio-mixing term of the governing equation and thus the numerical difficulty is easily compared between mixing styles by showing matrices represented by $(1 - \phi_i)K_{\theta,ij}$ at $(i, j)$ or transition matrices corrected for porosity.

Changes in manuscript (Page numbers/Line numbers):

We added explanations of transition matrices to Section 2.2.2 (P7/L201-203). A figure showing the porosity-corrected transition matrices for Fickian, homogeneous and LABS mixing are added, as they correspond to the bioturbational transport part of Jacobian matrix (P29, Fig. 1). We added a more detailed explanation of the differences in transition matrices and numerical implementations between mixing styles, referring to the new figure of the transition matrices (P8/L229-P9/L241).

General comment 5:

"Finally, the experiment descriptions also leave too many questions open. Here, especially the

species-dependent mixing experiment is poorly documented. The current text makes it impossible to understand how exactly the model has been set-up for this experiment."

Response:
We found questions raised for the experiments by the reviewer in specific comments 15 and 16. Please see our response to specific comments 15 and 16 by the reviewer.

Changes in manuscript (Page numbers/Line numbers):
Please see our changes in manuscript in response to specific comments 15 and 16 by the reviewer.

We provided additional descriptions of the experimental setups where we found deficient in the previous manuscript (e.g., P19/L549).

Specific comment 1:
"Page 8, Eq. (23): Equation (23) is only valid for intraphase biodiffusion; for interphase bioturbation, there is an extra term related to the porosity gradient (see, e.g. Munhoven, 2021)"

Response:
We agree. However, please note that the transition matrix method is flexible enough to enable implementation of interphase biodiffusion as well, based on the finite difference version of the governing equation (e.g., Munhoven, 2011). Also, the difference between intraphase vs. interphase biodiffusion is not a focus of this study, as we compare more drastically different bio-mixing styles: e.g., both itraphase and interphase biodiffusion are local mixing while we compare local vs. non-local mixing in this paper. Nonetheless, we agree that it is important to note that biodiffusion implemented in Eq. (25) (Eq. (23) in the previous manuscript) is intraphase biodiffusion.

Changes in manuscript (Page numbers/Line numbers):
We added sentences explaining that implemented biodiffusion is intraphse diffusion (P8/L211-213, P9/L255, P10/L262).

Specific comment 2:
"Similarly, the strengths and disadvantages of the different bioturbation model formulations are not discussed (the biodiffusive approach leads to block tridiagonal Jacobians, that can be inverted by an efficient block oriented Thomas algorithm, whereas the LABS derived transition matrix is likely to be full, without any special structure) and thus contributing to a Jacobian that will be computationally costly to invert."

Response:

Please see our response to general comment 4 by the reviewer where we address the issue.

Changes in manuscript (Page numbers/Line numbers):

Please see our changes in manuscript in response to general comment 4 by the reviewer.

Specific comment 3:

"Grid: The grid description is unfortunately not clear. It starts with a language problem: "[. . . ] discretized into $N = 100$ irregular grids where the grid size increases [. . . ]" does not make sense. Reading this as "[. . . ] discretized into $N = 100$ irregular grid intervals where the interval size increases [. . . ]", and using the information provided in Table 2 (mapping function and control parameter value) to generate the underlying distribution (with 101 grid points delimiting 100 grid intervals), and paying attention to avoid catastrophic numerical cancellation in the factor $(\beta - 1)$, with $\beta = 1 + 5 \cdot 10^{-11}$, one obtains a very odd result: the deepest grid interval takes more than 81% of the total extent (405.8 cm), the second deepest 2.86% (14.3 cm) and the third one 1.68% (8.41 cm). So, provided the reported information on the grid generation is accurate, I would consider the bottom of the grid as essentially useless. The resulting extreme numerical diffusion might possibly deteriorate the quality of all the results. The extreme jump in the grid interval lengths by a factor of almost 30 from the second deepest to the deepest interval (compared to less than 2 from the third to the second deepest layer) will lead to tremendous numerical diffusion, which might significantly influence the results further up in the sediment column. Although the adopted mapping function from the normalized regular to the finally used irregular grid is continuous (and continuously differentiable) – a condition that irregular grid mappings have to fulfil to ensure that consistency and convergence order of the numerical discretization schemes remain unchanged – the difference in the grid interval lengths acts, from a practical point of view more, like a discontinuity, which might ruin the convergence properties of the algorithm."

Response:

We agree to revise the sentence to be clearer on the grid structure. We define grid points as centers of grid cells/layers and $N$ is defined as the number of grid points/cells/layers.

We did not argue that the user must use the grid of the present study. Rather it is just a default setting. Please see, for example, line 256 in the previous manuscript where we stated that one can assume a different grid structure. The user can change the grid structure by changing the total sediment depth as well as the $\beta$ value with which the mapping function creates a different irregular (or even regular) grid.

The effect of numerical diffusion at deep depths in the default sediment grid in this study is already discussed in Section 2.4.2. We minimized the effect of numerical diffusion in deep sediment

layers on signal tracking by reading signals at the bottom of mixed layer, as explained in Section 2.4.2. This method should be effective to reduce the numerical diffusion regardless of the grid structure.

Numerical diffusion in deep sediment does not significantly affect diagenesis in upper column sediment in the simulations conducted in the present paper. As inferred from the reviewer's comment, most reactions occur within the mixed layer (e.g., Fig. 6) and thus introducing deep sediment layers to the model have little influences on overall $CaCO_3$ diagenesis (please find an exception in the paragraph just below, related to the reason why we adopted the grid including deep layers).

We included deep sediment layers in the simulations presented in this paper because it allows us to illustrate the effects of numerical diffusion on signal tracking and also the method to minimize its effects regardless of grid structure (Section 2.4.2). Furthermore, the model is designed to be applicable even to locations where deep reactions such as anoxic oxidation of methane influence $CaCO_3$ diagenesis (please also see our response to general comment 3 by the reviewer). For the above reasons, including deep layers is not completely useless.

As for numerical convergence, $CaCO_3$ systems (multiple classes of $CaCO_3$, ALK and DIC) are solved until the maximum relative concentration difference of all species at all depths become less than $10^{-6}$. This criterion has been satisfied in all simulations shown in this paper. Please also note that the model always checks the satisfaction of the governing equations by monitoring depth integrated fluxes of all relevant time change rate terms for all species. Namely, for each simulated species, fluxes caused by amount change in sediment (cf. 'non-steady-state' flux; Kanzaki et al., 2019), advection, diffusion, bio-mixing, raining, individual reactions, and so on, as well as the residual of all the above fluxes (which should ideally always be zero). The residual fluxes have always been negligible (e.g., $<10^{-6}$ of the imposed rain fluxes) for all the species in all the simulations and thus we consider that the governing equations are satisfactorily solved with negligible errors even with the irregular grid used in this paper.

To demonstrate that the grid structure can be flexible in IMP and does not affect the results, we repeated the same dissolution experiments as in Section 3.2.2 but with adopting a shallower ($z_{tot} = 50$ cm) and less irregular ($\beta = 1.05$) grid in the Supplementary material (Section S1.4). Please note that the grid becomes more irregular when $\beta$ is closer to 1 and compare with the default grid in the manuscript ($z_{tot} = 500$ cm and $\beta = 1+5\times10^{-11}$). The obtained depth profiles as well as tracked proxy signals are very similar to the results in Section 3.2.2, supporting that numerical convergence is satisfied with the default grid and also that the proxy tracking scheme in Section 2.4.2 works regardless of the grid structure.

Changes in manuscript (Page numbers/Line numbers):

The sentence relevant to the definition of $N$ was revised (P10/L269-271).

We added an example of using a different grid structure to the Supplementary material (Sections

S1.2 and S1.4). We referred to the Supplementary material where we state that the user can adopt different grid structures (P10/L271-273).

We added explanations on convergence criteria and the satisfaction of the governing equations in all the simulations to Section 2.3.2 (P11/L322, P12/L329-333).

Specific comment 4:
"One might furthermore wonder why the model grid has to extend to a depth of 5 m. The useful ("undisturbed" information is tapped at a much shallower depth, just below the mixed layer (i.e., typically at 10–20 cm depth). With a calcite/clay input mass ratio of about 9:1, at most of the order of 10 cm of sub-mixed-layer sediment can possibly be unburied during a chemical erosion event and the information stored in the gridded part of the sediment column deeper than, say 20–30 cm is actually not required here.."

Response:
Please see our response to specific comment 3 by the reviewer where we address the issue mostly.

We agree with the reviewer that most reactions occur within 10–20 cm depths. Nonetheless, the capability to adopt different grid structure increases the applicability of the model even to specific sites such as those where DIC and ALK are significantly produced from deep AOM (e.g., Bradbury and Turchyn, 2019, Earth Planet. Sci. Lett. 519, 40). Please also note that we did not argue that the model must assume the specific grid structure adopted in this paper; rather we stated that the user can adopt a different grid structure.

Changes in manuscript (Page numbers/Line numbers):
Please see our changes in manuscript in response to specific comment 3 by the reviewer.

Specific comment 5:
"I finally also wonder whether the recorded "undisturbed" signal would be consistent with the signal recorded in the main sediment column, even if there were no numerical diffusion – numerical diffusion was put forward as a reason for tapping the useful information right below the mixed layer: even in the absence of numerical diffusion, the signal in the main column may be altered by sub-mixed-layer chemical reactions. Furthermore, if chemically altered material later becomes subject to chemical erosion (i.e., returns to the mixed-layer), its composition can be expected to be different from that in the "undisturbed" record. How important might the resulting inconsistencies be?"

Response:
As commented by the reviewer (please see our response to specific comment 4 by the reviewer),

most of the reactions (at lease those simulated in the main text) occur within the mixed layer. Nonetheless, the model can read and record proxy signals at a different depth point than the bottom of mixed layer by specifying the reading point within the code. The flexibility of IMP to use different grid structures (e.g., modifying $z_{tot}$, $N$ and $\beta$) and reading points will allow the model to deal with the case when chemical reactions propagate below the mixed layer.

Changes in manuscript (Page numbers/Line numbers):
We mentioned that the flexibility of the depth point where signals are read in the revised manuscript (P14/L396-399).

Specific comment 6:
"Page 10, lines 267–273: Why not move this transition matrix representation of the biodiffusive model of bioturbation to Sect. 2.2.2 ("Bioturbation")? In that place it would contribute to emphasize the overarching nature of the transition matrix approach. A priori, it was not clear in my mind that biodiffusion could also be part of the transition matrix approach (albeit in discretized form)."

Response:
Agreed.

Changes in manuscript (Page numbers/Line numbers):
Revised as suggested (P8/L204-P8/L228).

Specific comment 7:
"Section 2.5.1 (pages 10–11): This is obviously the central part of this manuscript. I find it quite short in that respect. The advantages/disadvantages (potentials/shortcomings) of the different methods could be stated in more detail. I would also like to challenge the authors on several assertions made here. I do not see why method 3 is less flexible than methods 1 and 2. To me, methods 2 and 3 are mathematically speaking absolutely identical. The only difference I can see is in the way the input data have to be processed. That processing can, however, be considered to be carried out outside the model itself."

Response:
Please see our response to specific comment 8 by the reviewer where we address the issue about advantages/disadvantages of methods 1 and 2.
     The processing of input proxy signals is closely linked to how we define $CaCO_3$ classes to be simulated within sediment and their rain fluxes. Without this procedure, the model cannot do signal

tracking diagenesis (Sections 3.2 and 3.3) and thus is a crucial part of the model.

We agree that method 3 can be regarded as a derivative of method 2 as stated in line 364. However, method 3 is not completely the same as method 2. Method 3 in the current version tracks $\delta^{13}C$, $\delta^{18}O$, $\Delta_{47}$ and $^{14}C$ age using 5 $CaCO_3$ classes that correspond to 5 isotopologues: $Ca^{12}C^{16}O_3$, $Ca^{12}C^{18}O^{16}O_2$, $Ca^{13}C^{16}O_3$, $Ca^{13}C^{18}O^{16}O_2$, and $Ca^{14}CO_3$ (Section 3.3 and Section S1.1.2 in the Supplementary material). On the other hand, method 2 can track $\delta^{13}C$ and $\delta^{18}O$ with 4 $CaCO_3$ classes that possess unique combinations of the maximum and/or minimum input signal values (e.g., Sections 3.2.1 and 3.2.3). We can evaluate the flexibility of the two methods by considering the case where we attempt to track $\delta^{11}B$ instead of $\delta^{18}O$. It is not possible to track $\delta^{11}B$ with the above 5 classes of method 3 because $\delta^{11}B$ cannot be defined by any combinations of the above 5 isotopologues. However, it is possible to track $\delta^{11}B$ with method 2 because one only has to use input values of $\delta^{11}B$ as those of $\delta^{18}O$ and obtained signal records can be regarded as those for $\delta^{11}B$. Thus method 2 is more flexible than method 3.

Changes in manuscript (Page numbers/Line numbers):

Please see our changes in manuscript in response to specific comment 8 by the reviewer.

We added more explanations on method 3 (P13/L366-367, P13/L370-371).

We also added a section conducting experiments using method 3 (Section 3.3, P20/L601-P22/L646), which we believe will help the reader better understand the difference of method 3 from method 2.

Specific comment 8:

"Page 12, lines 332–345: The discussion about how time tracking could be implemented is, as far as I can see, not entirely correct. Obviously, method 1 would be extremely costly to adopt for time, but it clearly is the one that would offer the best time resolution. Method 1 essentially discretizes the complete age-dimension carried by one sedimentary component (e.g., calcite) as an ever-growing number of $CaCO_3$ variants, each one representing one age-class. I is difficult to comprehend how method 2 could possibly conserve more accurate information with only two variables, compared to method 1. For any given time-step, the input data for method 1 are given by one single class, and those for method 2 by one interpolated value between the end members. To me, one appears as accurate as the other, for any given single time-step. For the next time step, another single class is created and added for method 1, while another interpolated value between the end members is added for method 2, and so forth. After n time steps with method 1, one can clearly distinguish between the fates of each single class of material brought in during the n steps; with method 2, this is not possible, as the n interpolates have been added, with little chance to deconvolve the resulting information. So, to me, method 1 is the accurate one, not method 2."

Response:

We agree with the reviewer that "one appears as accurate as the other". This has been actually confirmed (Supplementary material). As a useful feature of IMP, one can compare the results of three signal tracking methods. When tracking $\delta^{13}$C and $\delta^{18}$O, the three methods yield essentially the same results, although the accuracy of method 1 compared to the other methods depends on the number of tracked $CaCO_3$ classes or the resolution with which model time is discretized, confirming the disadvantage of method 1 described in the manuscript and by the reviewer. Indistinguishable results between methods 2 and 3 suggests that the results with method 2 should be consistent with those from the traditional models of signal tracking diagenesis (e.g., Keir, 1984) where a direct tracking method is adopted.

The argument by the reviewer that particle information tends to be lost or uncaptured in method 2 is not necessarily the case. For instance, when one tracks a proxy in $CaCO_3$ particles that belong to a single model species with the common physicochemical properties (size etc.), then method 2 does not miss any information relative to method 1. More generally, method 2 does not miss any information relative to method 1, as long as physicochemical properties that change with proxy signals are tracked. Please note that a physicochemical property of $CaCO_3$ particles (e.g., size) can be tracked as a proxy by either method. With method 1, each class of $CaCO_3$ particles can possess any number of proxies as well as physicochemical properties (an advantage; please see, however, the numerical disadvantage of the method above). With method 2, one has to multiply the number of $CaCO_3$ classes by 2 per addition of one proxy or one physicochemical property to be tracked so that each $CaCO_3$ class has a unique combination of end-member proxy values and physicochemical properties (a potential numerical disadvantage; please also see our response to minor/technical correction 15 by the reviewer).

As such, methods 1 and 2 are both accurate if an infinite computational power would be available. However, the computational disadvantage of method 1 tends to overwhelm because a fine time resolution is needed to obtain 'good' results especially when time-dependent diagenesis is simulated (please also see our response to specific comment 10 by the reviewer where we address the issue about the time steps adopted by the model).

Changes in manuscript (Page numbers/Line numbers):

We added more explanation to Section 2.4.1 (P12/L344, P12/L353-P13/L357, P13/L376-378).

We added Tables 4 and 5 where we list the properties of $CaCO_3$ classes used for the experiments in Sections 3.2.3 and 3.3 to make it easier for the reader to understand how method 2 or 3 can track particle properties (P48).

Please also see our changes in manuscript in response to specific comment 10 and minor/technical correction 15 by the reviewer.

Specific comment 9:

"Page 14, lines 410–412: "This difference can be explained by a burial velocity enhancement caused by high organic matter preservation in the oxiconly model, which is not considered by Archer (1991)." This is probably correct, but why not check it instead of speculating? This can be easily done by setting $V_{OM}$ to 0."

Response:

We confirmed our argument by conducting lysocline experiments with $V_{OM} = 0$.

Changes in manuscript (Page numbers/Line numbers):

We added the results of the lysocline experiments to the Supplementary material (Section S2.1) and referred to them in the relevant sentence (P16/L480).

Specific comment 10:

"Throughout: there seems to be a mismatch between "time step" and "instants in time" Actually, no information about the step lengths (time steps) adopted for the integration of Eq. (1) in time."

Response:

We agree that little information on time steps was provided in the previous manuscript.

Changes in manuscript (Page numbers/Line numbers):

We provided information on time steps (P12/L324-328). We also corrected where "time step" and "instants in time" are misused (e.g., P15/L432)

Specific comment 11:

"Page 14, lines 398–400: "Absence of significant void spaces or expansions in solid sediment has been adopted as a convergence diagnostic by the sediment diagenesis model of Archer et al. (2002), [. . . ]" This sentence could possibly be misunderstood, as this is actually not the only diagnostic used by Archer et al. (2002). Their convergence diagnostic is first "[. . . ] based on conservation of mass to within 2% for all solid phase and dissolved constituents (appropriate balances between rain, reaction, diffusion, and burial rates)." (Archer et al., 2002, p. 17-4, par. [19]). They continue writing that the sum of the solid phase concentrations also provides a convergence diagnostic."

Response:

We did not state or argue that Archer et al. (2002) adopted the sum of solid phase concentrations as the only diagnostic. Nonetheless, we agree with the reviewer that the relevant sentence might cause a

potential misunderstanding of the model by Archer et al. (2002) and thus agree to revise the sentence.

Changes in manuscript (Page numbers/Line numbers):
We revised the sentence to avoid potential misunderstanding (P16/L466-468).

Specific comment 12:
"Conserving mass to within a few % only – a side-effect of requiring letting the sum of the solids' volume fractions float within a few % (line 255, page 9)) – might be sufficient for steady-state calculations. In transient simulation experiments, I would anticipate that deviations of the order of a few % could cause considerable model drift in transient simulation experiments. It would be better not to leave any loose ends and therefore to enforce strict static volume conservation."

Response:
In the updated version of the model (v1.0), we enabled the calculation scheme of clay concentration by Munhoven (2021), which is based on the volume conservation Eq. (22) and thus with which the total solid volume fraction diverges only negligibly from 1. The calculation with this new scheme yields essentially the same results as those with the previous version (v0.9) for the experiments conducted in this study. We changed the default method of clay calculation to the above scheme although the clay calculation scheme used in v0.9 (i.e., solving Eq. (1) directly) still remains in the code and can be chosen. Accordingly, all figures are reproduced (Figs. 6-16) with the new clay calculation scheme.

Changes in manuscript (Page numbers/Line numbers):
We changed the default calculation scheme of clay concentration and its description (P11/L305-307). Figure 6-15 were accordingly all revised (P34-43).

Specific comment 13:
"The statement at line 400 ("The results of the second experiment thus confirm [. . . ]") is a non sequitur. The results do not confirm the applicability of the model for time-dependent simulation experiments: they only do not infirm it."

Response:
We agree.

Changes in manuscript (Page numbers/Line numbers):
We deleted the sentence.

Specific comment 14:

"Page 15, lines 437ff : Although I understand that multiplying the transition matrix by 1/10 might facilitate the calculation, it also reduces the mixing intensity by a factor 10, and thus describes a completely different setting. Are the results for the so reduced mixing intensity accepted as final in case the calculations converge, or are they used as a starting point for a continuation method, wherein the mixing intensity is then gradually increased to approach the original matrix? I think that the results obtained with the matrix divided by 10 cannot be used for a comparison with others where no such reduction was adopted."

Response:

The original LABS transition matrix was obtained in a LABS simulation where a biodiffusion coefficient was estimated to range from 0.1 to 10 $cm^2$ $yr^{-1}$ based on particle displacements (cf., Kanzaki et al., 2019). Therefore, a factor of 1/10 is likely reasonable as it makes LABS mixing comparable with Fickian mixing (0.15 $cm^2$ $yr^{-1}$). We did not change transition matrices during a simulation.

Changes in manuscript (Page numbers/Line numbers):

We added more information on LABS mixing adopted in this paper (P8/L218-228). Please also find that we added a figure to compare transition matrices corrected for porosity (corresponding to Jacobian matrices) between different mixing styles in response to general comment 4 by the reviewer (P29).

Specific comment 15:

"Section 3.2.3, pages 17–18 This experiment needs to be better documented. Critical information is missing here: what bioturbation model has been used to get these results? In the text, we read about different bioturbation depths for small and large particles, with reference to Bard (2001). The bioturbation model of Bard (2001) is, mathematically speaking, rather convoluted and it is not clear to me how it might be transposed to the transition matrix framework adopted here. At first sight, it is possibly a homogeneous mixing model, but I am not entirely even sure that it is possible to transpose it at all into the transition matrix framework of IMP. In that model, the bioturbation depth is the main control parameter. How that bioturbation depth has been used as a control parameter here – if it has, which is unclear as well – is not explained. Please make this description more complete and if necessary also provide additional theoretical foundations"

Response:

We did not use the mixing model of Bard (2001) but referred to the study by Bard (2001) because we adopted different mixing depths for different size classes of $CaCO_3$ as in Bard (2001). We used Fickian and homogeneous mixing as defined by Eqs. (18) and (19). For the fine species these equations also reflect the change in mixed layer depths (i.e. to 20cm).

Mixed layer depth is one of the parameters that control bio-mixing and the transition matrices as explicitly formulated in Eqs. (18) and (19) in Section 2.2.2. We provided the depths of mixed layers for fine and coarse species, which are sufficient to create the transition matrices for Fickian and homogeneous mixing, with all other parameters remaining the same based on Eqs. (18) and (19). We thus do not consider additional theoretical foundations are necessary. We agree to provide more details to avoid potential confusion.

Changes in manuscript (Page numbers/Line numbers):
We added more descriptions regarding the bio-mixing adopted in the experiment in Section 3.2.3 (P20/L580-582, P20/L584).

Also we added a table (Table 4) so that it is easier to understand the properties of $CaCO_3$ classes used for the experiment in Section 3.2.3 (P48).

Specific comment 16:
"It should also be noticed that some theoretical homework is of order when it comes to different mixing rates for small and large particles. The solids' advection rate is actually always defined as the movement rate of the bulk solids (Meysman et al., 2005, see, e.g.). Accordingly, the advective rate may not be equal to w in Eq. (1) – this is already not the case when interphase biodiffusion is adopted as a bioturbation model. In case a biodiffusion model is used, it should be noticed that Eq. (23) is not applicable (this is mentioned as a sufficient condition at lines 214–216 for Eq. (23) to hold, but it is also necessary)."

Response:
We consider that we have provided enough theoretical framework to calculate $w$ even for a simulation that enables different mixing styles between solid species. As formulated in Eq. (21), $w$ is defined as moving velocity of bulk solids as in other models including Munhoven (2021), i.e., all solid species are advected with the rate $w$. The $w$ profile is calculated according to Eq. (21) accounting for the volume changes of all solid species including multiple $CaCO_3$ classes caused by reactions and mixing (local or non-local). Thus changing mixing styles between solid species does not change the calculation scheme of $w$, that is essentially based on Eq. (21). Please find that applicability of Eq. (25) (Eq. (23) in the previous manuscript referred to by the reviewer), a simplification of Eq. (21), is quite limited, only when all solid species are bio-mixed in intraphase diffusion with the same mixing depths and biodiffusion coefficients for all solid species as stated in

lines 255-264. For example, introducing a single solid species characterized by a different mixed layer depth to the above case makes Eq. (25) inapplicable and Eq. (21) has to be used instead. This ensures that the different solid volume distribution of the species caused by mixing to a different mixed layer depth is correctly reflected in the calculation of solid-phase advection-rate profile.

We agree that when we adopt the interphase biodiffusion for bio-mixing, Eq. (25) is not applicable. In that case, one has to adopt Eq. (21) to calculate $w$ using the corresponding transition matrix.

Changes in manuscript (Page numbers/Line numbers):
We emphasized that Eq. (25) is only applicable when bio-mixing is intraphase biodiffusion with the same mixing intensity and depth for all the solid species (P9/L255, P10/L262).

We also stated that including interphase biodiffusion requires a different transition matrix from that created by Eq. (18) (P8/L213).

Technical and minor correction 1:
"Abstract: Please include the meaning of the model name acronym somewhere in the abstract."

Response:
We agree to put the meaning of the model name acronym in the abstract.

Changes in manuscript (Page numbers/Line numbers):
We put the full model name in the abstract (P1/L10-11).

Technical and minor correction 2:
"Page 3, lines 65–66: "The reason for this is that published sediment mixing models are generally unable to realize diagenetic reaction" – not sure what is meant here (should "are generally unable to realize" possibly read "do generally not take into account"?)"

Response:
Agreed.

Changes in manuscript (Page numbers/Line numbers):
Corrected as suggested (P3/L65-66).

Technical and minor correction 3:

"Page 3, line 81: "Following presentation" should read "Following the presentation" (or "After the presentation" ?)"

Response:
Agreed.

Changes in manuscript (Page numbers/Line numbers):
Corrected as suggested (P3/L81).

Technical and minor correction 4:
"Page 4, line 91: Is it possible to "share" distinct characteristics?"

Response:
We meant 'possess the common, distinct characteristics'.

Changes in manuscript (Page numbers/Line numbers):
We corrected the sentence in accord with our response above (P4/L92).

Technical and minor correction 5:
"Page 6, line 160: "mocsy" should read "mocsy 2.0""

Response:
We thank the reviewer for pointing this out.

Changes in manuscript (Page numbers/Line numbers):
Corrected as suggested (P6/L165).

Technical and minor correction 6:
"Pages 6 and 7: The notation $dz_i$ for the thickness of layer i is not recommendable, as there are later ratios of such thicknesses are used (e.g., page 7, Eqs. (13) and (14)), which could be confused with derivatives and thus lead to unnecessary misunderstandings. I suggest to replace $dz_i$ by $\delta_i$ or $\Delta_i$, perhaps hi."

Response:
Agreed.

Changes in manuscript (Page numbers/Line numbers):
We changed $dz_i$ to $\delta z_i$ (P7/L184-185, P7/L187, P7/L190, P7/L195).

Technical and minor correction 7:
"Page 7, line 181: Is $A$ the horizontal cross-sectional area?"

Response:
Yes.

Changes in manuscript (Page numbers/Line numbers):
Changed the description of $A$ (P7/L186).

Technical and minor correction 8:
"Page 7, lines 195–196: Very cumbersome and syntactically incorrect sentence. I suggest to reformulate it as "Equation (15) is a finite difference version of Eq. (17) . . . """

Response:
We agree to revise as suggested.

Changes in manuscript (Page numbers/Line numbers):
Revised as suggested (P7/L200).

Technical and minor correction 9:
"Page 8, lines 212 and 213: I suggest to replace $D_b$ by $D_{\theta,b}$ or something the like in htese two instances to emphasize that different biodiffusion coefficient values may be used for different particle classes (different classes of particles are later supposed to be transported in different ways by bioturbation) The special case for a single $D_b$ for all solids then comes more naturally at line 214."

Response:
Agreed.

Changes in manuscript (Page numbers/Line numbers):
Corrected as suggested (P8/L207, P8/L209-210, P9/257-258).

Technical and minor correction 10:

"Page 9, lines 236–237: This sentence may be misleading, as "specified at the beginning of each time integration" could be wrongly interpreted as saying that an time-explicit approach is used in IMP"

Response:

We agree to revise the relevant sentence to avoid potential confusion.

Changes in manuscript (Page numbers/Line numbers):

We revised the sentence (P10/L284).

Technical and minor correction 11:

"Page 9, lines 253: How is "significantly different" translated quantitatively?"

Response:

Burial velocity is regarded to be converged when the relative difference becomes less than $10^{-6}$ from the previous iteration. When this criterion is not met within 20 iterations (only encountered in a few conditions in lysocline experiments), the solution with minimum relative difference is adopted (still less than a few %).

Changes in manuscript (Page numbers/Line numbers):

Description such as above was added to Section 2.3.2 (P11/L309-313).

Technical and minor correction 12:

"Page 10, lines 277ff : Wich version of LABS was used here in the end? Reed et al. (2007)? Or was it eLABS (Kanzaki et al., 2019)? Please specify."

Response:

We used the eLABS code by Kanzaki et al. (2019), but we disabled the new functionalities in eLABS (e.g., 2D reactive-transport of oxygen and OM as well as fluid flow) to obtain a transition matrix controlled dominantly by biological parameters.

Changes in manuscript (Page numbers/Line numbers):

The description such as above was added to Section 2.2.2 (P8/L218-221).

Technical and minor correction 13:

"Page 10, Equation system (24): First of all, this way of defining ($K_{\theta,ij}$) is difficult to understand. At first it looks like some kind of implicit definition. Is there not more clear way to write this? Second there seem to be two errors:

• "$2 \leq j = i + 1 = n_{ml}$" should probably read "$2 \leq j = i + 1 \leq n_{ml}$"

• "$1 \leq j = i - 1 = n_{ml} - 1$" should probably read "$1 \leq j = i - 1 \leq n_{ml} - 1$""

Response:

We thank the reviewer for pointing out the typos, which were made when the first author changed the manuscript format from Microsoft Word to LaTex.

Changes in manuscript (Page numbers/Line numbers):
Corrected as suggested (P8/L209).

Technical and minor correction 14:

"Page 11, line 309: "[. . . ] and accompanied generation of alkalinity, [. . . ]": not sure what this could mean. With $Ca^{14}CO_3/CaCO_3$ ratios of the order of $10^{-14}$, alkalinity changes by $Ca^{14}CO_3$ decay should really be on the negligible side of life."

Response:

Alkalinity can be produced assuming carbonate $^{14}C$ is decayed into nitrate $^{14}N$. We agree that alkalinity production by radiocarbon decomposition is minute.

Changes in manuscript (Page numbers/Line numbers):
We added more explanations such as above to Supplementary material where method 3 (the direct tracking method) is detailed.

Technical and minor correction 15:

"Page 11, lines 294 and 298: "$2^{n_p}$" should read "$2n_p$" as each proxy requires two end-members."

Response:

With method 2 (interpolation method) for signal tracking, each $CaCO_3$ class must have a unique combination of end-member proxy values. In other words, the number of $CaCO_3$ classes required to track $n_p$ proxies is identical to the number of possible unique combinations of endmember proxy values. Because one class of $CaCO_3$ can have either of 2 end-member values for a given proxy, the total number of possible unique combinations of end-member values for $n_p$ proxies is give by

$$\overset{\text{proxy 1}}{2} \times \overset{\text{proxy 2}}{2} \times \cdots \times \overset{\text{proxy } n_p}{2} = 2^{n_p}$$

Therefore, method 2 requires $2^{n_p}$ $CaCO_3$ classes for tracking $n_p$ proxies.

As an example, the experiment in Section 3.2.3 tracks 2 proxy and 1 physical property (i.e., size) and requires 8 classes of $CaCO_3$.

Changes in manuscript (Page numbers/Line numbers):
We added more explanations such as above to the relevant sentences (P12/L354-P13/L357).

We also added 2 tables that tabulate the properties of $CaCO_3$ classes in simulations in Sections 3.2.3 and 3.3 to show that adding 1 proxy/property to be tracked necessitates an increase of the number of $CaCO_3$ classes by a factor of 2 as in the above equation (P48).

Technical and minor correction 16:
"Page 13, lines 364–365: "5 time steps" should most probably read "five instants in time". By the way, which time step was chosen for the integration? A variable one? a constant one – how long?"

Response:
We agree to correct the sentence.

The time step can be specified by the user. In the default setting, it was dependent on the phase of experiment: during spin-up phase to the initial steady state, it increases with model time, from 100 to $10^5$ yr; then, time step is either 5 or 10 yr for a 10 kyr or 50 kyr signal change event, respectively; and the time step afterwards to reach another steady state is the same as that in the signal change event.

Changes in manuscript (Page numbers/Line numbers):
The sentence was corrected as suggested (P15/L432).

We added explanations such as above to Section 2.3.2 (P12/L324-328).

Technical and minor correction 17:
"Page 13, lines 377ff : It would be fair to state that these are replications of experiments from Archer (1991)."

Response:
Agreed.

Changes in manuscript (Page numbers/Line numbers):

We added "(cf. Archer, 1991)" to line 441 (P15/L441).

Technical and minor correction 18:
"Page 14, line 392–393: Strange sentence. – please reformulate."

Response:
Agreed.

Changes in manuscript (Page numbers/Line numbers):
We revised the sentence (P16/L460-461).

Technical and minor correction 19:
"Page 14, line 410: "than Archer (1991) model" would more correctly read "than the model of Archer (1991)""

Response:
Agreed.

Changes in manuscript (Page numbers/Line numbers):
Revised as suggested (P16/L478).

Technical and minor correction 20:
"Page 14, line 434: "provability" should probably read "probability""

Response:
We are grateful to the reviewer for pointing out the typo.

Changes in manuscript (Page numbers/Line numbers):
We corrected the typo as suggested (P17/L501).

Technical and minor correction 21:
"Page 14, line 413: "in $CaCO_3$ rain" should read "of the $CaCO_3$ rain""

Response:

Agreed.

Changes in manuscript (Page numbers/Line numbers):
Revised as suggested (P16/L482).

Technical and minor correction 22:
"Page 15, line 439: "are now shown" should read "are not shown", I guess."

Response:
Apologies for many typos and we are grateful to the reviewer for pointing them out.

Changes in manuscript (Page numbers/Line numbers):
We corrected the typo as suggested (P17/L503).

Technical and minor correction 23:
"Page 16, line 455: would "at depths" not better read "from depths"?"

Response:
Agreed. But we removed the sentence in the revised manuscript.

Changes in manuscript (Page numbers/Line numbers):
The sentence was removed.

Technical and minor correction 24:
"Page 16, lines 470 and 472: Text imprecise: chemical erosion requires dissolution, but dissolution does not necessarily lead to chemical erosion. Please reformulate."

Response:
Agreed.

Changes in manuscript (Page numbers/Line numbers):
We changed 'chemical erosion' to 'significant dissolution' in the sentence (P18/L534).

Technical and minor correction 25:

"Page 17, line 490: "When dissolution is imposed [. . . ]"? Would "When dissolution is intensified [. . . ]" not be more correct?"

Response:
Agreed.

Changes in manuscript (Page numbers/Line numbers):
Revised as suggested (P19/L555).

Technical and minor correction 26:
"Page 17, lines 503–504: "Simulated proxy signals are considerably shorter in apparent duration as described in the above paragraph." – not sure what this means."

Response:
We meant that signal changes are recognized in shorter depth intervals.

Changes in manuscript (Page numbers/Line numbers):
Corrected in accord with the above response (P19/L568-569).

Technical and minor correction 27:
"Page 18, line 520: "the more" should read "the better""

Response:
Agreed.

Changes in manuscript (Page numbers/Line numbers):
Revised as suggested (P20/L591).

Technical and minor correction 28:
"Page 18, line 521: "accumulation rate differs between" would better read "accumulation rates are different for""

Response:
Agreed.

Changes in manuscript (Page numbers/Line numbers):
Revised as suggested (P20/L592).

Technical and minor correction 29:
"Page 18, line 539: "The source codes of IMP model" should read "The IMP source codes" (delete "model")"

Response:
Agreed.

Changes in manuscript (Page numbers/Line numbers):
Revised as suggested (P22/L660).

Technical and minor correction 30:
"Page 18, line 540: "specific version used of the model" should read "specific model version used""

Response:
Agreed.

Changes in manuscript (Page numbers/Line numbers):
Revised as suggested (P22/L661).

Technical and minor correction 31:
"Page 22, lines 636–637: This URL points to the secondary JSTOR archive copy of the reference. It better had to be replaced by the DOI of the original paper (available in open access): DOI:10.5670/oceanog.2009.100"

Response:
Agreed. We are grateful to the reviewer for pointing this out.

Changes in manuscript (Page numbers/Line numbers):
We revised the DOI of the referred paper (P26/L763-764).

Technical and minor correction 32:

"Page 24, line 710: The DOI of the MATLAB version 1.1 of CO2SYS is not resolving any more. The current URL is https://cdiac.ess-dive.lbl.gov/ftp/co2sys/CO2SYS_calc_MATLAB_v1.1."

Response:
We are grateful to the reviewer for pointing this out.

Changes in manuscript (Page numbers/Line numbers):
The URL of CO2SYS has been updated (P28/L845).

Technical and minor correction 33:
"Page 39, Table 1:

- I guess, "Number of sediment grids" means "Number of sediment grid points" as there is most probably only one grid.
- For the density of OM, a value of 1.2 $g/cm^3$ is reported with reference to Mayer et al. (2004). I have not been able to find that value of 1.2 $g/cm^3$ in Mayer et al. (2004). Considering the $\rho_{OM}$ values reported for marine samples in Table 1 of that reference, I find a higher value of $1.45 \pm 0.23$ $g/cm^3$. Please clarify.

It would be good to specify more clearly that OM $\equiv$ $CH_2O$. Only in this case some of the ratios such as the OM:$CaCO_3$ ratio $r$ make sense, as one mole of OM then represents one mole of OC. Readers used to Redfield composition might be confused else."

Response:
We agree on the first point.

Mayer et al. (2004) did not argue that the middle value of their reported range (1.14–1.68 g cm$^{-3}$) is the most likely density of sediment OM. We have taken a value close to the lower limit of their reported range because choosing a relatively high value might not be able to reproduce a wide range of sediment density, which can be as low as 1.2 g cm$^{-3}$ (e.g., Hamilton, 1976, J. Sediment. Res., 46, 280). Also, 1.2 g cm$^{-3}$ is not necessarily unreasonably low value. For instance, other models take similar values (e.g., 1.0 g cm$^{-3}$ by Meyers, 2007).

We agree to clarify that OM $\equiv$ $CH_2O$.

Changes in manuscript (Page numbers/Line numbers):
We revised Table 1 in accord with the above response (P45).

We clarified OM $\equiv$ $CH_2O$ at the beginning of Section 2.2 (P4/L114).

Technical and minor correction 34:

"Figures 2, 7, 8, 10: It is recommended not to use green and red/orange colours tones in parallel on a graph (see https://www.geoscientific-modeldevelopment.net/submission.html - "Figures & Tables", point 7)"

Response:
We are grateful to the reviewer for pointing this out to us and useful URL.

Changes in manuscript (Page numbers/Line numbers):
We changed colors in Figs. 3, 8, 9 and 11 (Figs. 2, 7, 8 and 10 in the previous manuscript) (P31, P36, P37, P39). Other figures are also modified using colorblind safe colors (cf. https://personal.sron.nl/~pault/) (P34-35, P38, P40-43).